# Catalyst switch strategy enabled a single polymer with five different crystalline phases

Pengfei Zhang ®[1], Viko Ladelta ®[1], Edy Abou-hamad[2], Alejandro J. Müller[3] & Nikos Hadjichristidis ®[1] ✉

Well-defined multicrystalline multiblock polymers are essential model polymers for advancing crystallization physics, phase separation, self-assembly, and improving the mechanical properties of materials. However, due to different chain properties and incompatible synthetic methodologies, multicrystalline multiblock polymers with more than two crystallites are rarely reported. Herein, by combining polyhomologation, ring-opening polymerization, and catalyst switch strategy, we synthesized a pentacrystalline pentablock quintopolymer, polyethylene-*b*-poly(ethylene oxide)-*b*-poly(ε-caprolactone)-*b*-poly(L-lactide)-*b*-polyglycolide (PE-*b*-PEO-*b*-PCL-*b*-PLLA-*b*-PGA). The fluoroalcohol-assisted catalyst switch enables the successful incorporation of a high melting point polyglycolide block into the complex multiblock structure. Solid-state nuclear magnetic resonance spectroscopy, X-ray diffraction, and differential scanning calorimetry revealed the existence of five different crystalline phases.

Semi-crystalline polymers have a wide range of applications in self-assembly, self-healing, and special engineering materials[1–5]. The crystalline phase of such polymers contributes to superior mechanical properties[6], high thermal conductivity[7], and better phase separation[8]. The most well-known semi-crystalline polymer is polyethylene (PE), which is widely used in packaging, agriculture, engineering, etc[9,10]. Other typical examples of semi-crystalline polymers are poly(ethylene oxide) (PEO), poly(ε-caprolactone) (PCL), and poly(L-lactide) (PLLA). They are mostly synthesized by anionic ring-opening polymerization (ROP), affording various well-defined semi-crystalline polymers with controllable molecular weight and low dispersity (Đ).

The recent advances in polymer synthesis have facilitated the development of double- and multi-crystalline block polymers[11,12]. The most common double-crystalline diblock polymers prepared by ROP are PEO-*b*-PCL[13–15], PEO-*b*-PLLA[16–18], and PCL-*b*-PLLA[19–22], where the common PE-based double-crystalline polymers are PE-*b*-PEO[23–25], PE-*b*-PCL[26,27], and PE-*b*-PLLA[28,29]. The PE precursor (ω-hydroxyl-terminated PE, PE-OH) is conventionally prepared by anionic polymerization of 1,3-butadiene followed by end-capping with one unit of ethylene oxide to afford ω-hydroxyl-terminated polybutadiene (PB-OH), followed by hydrogenation with the Wilkinson's catalyst. However, PE-OH prepared by this method has unavoidable ~7% side chains (1,2-addition units)[30,31]. On the other hand, polyhomologation (or C1 polymerization) has emerged as an alternative method for preparing linear hydroxyl functionalized polyethylene using organoborane as the initiator[32–34]. One methylene group is inserted into the polymeric chain at a time during propagation. The resulting three-arm polyethylene with boron junction is oxidized/hydrolyzed to afford well-defined and perfectly linear PE-OH. This PE-OH can then be used as the macroinitiator for ROP, or combined with other polymerization techniques (e.g., atom transfer radical polymerization or iodine transfer polymerization)[35–38].

The interplay between crystallization and phase separation is extremely complex in multicrystalline polymers with more than three different crystalline domains and is one of the core subjects in polymer physics (Fig. 1)[39]. As the type of crystalline blocks increases, the nucleation, crystallization, and morphology become more complicated, unpredictable but more attractive as well. At the same time, the

[1]Polymer Synthesis Laboratory, KAUST Catalysis Center, Chemistry Program, Physical Sciences and Engineering Division, King Abdullah University of Science and Technology (KAUST), Thuwal 23955, Saudi Arabia. [2]Imaging and Characterization Core Lab, King Abdullah University of Science and Technology (KAUST), Thuwal 23955, Saudi Arabia. [3]Department of Polymers and Advanced Materials, Physics, Chemistry and Technology, Faculty of Chemistry, University of the Basque Country UPV/EHU, Paseo Manuel de Lardizabal 3, 20018 Donostia-San Sebastián, Spain. ✉e-mail: nikolaos.hadjichristidis@kaust.edu.sa

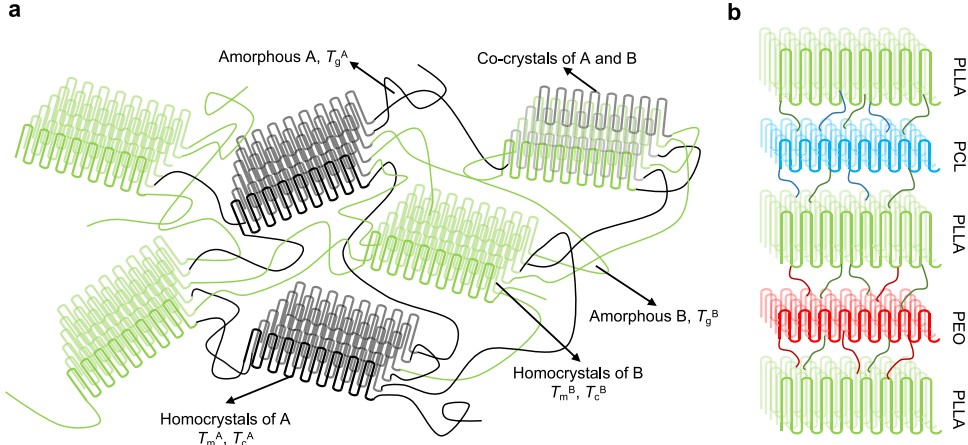

**Fig. 1 | The complex interplay of crystalline and amorphous phases in multiblock multicrystalline materials. a** Possible phases in a double crystalline diblock copolymer. **b** Schematic representation of a triple crystalline triblock terpolymer PEO-*b*-PCL-*b*-PLLA.

synthetic difficulty increases significantly because the polymerization method must be compatible with each block, demanding a strict requirement for the catalysis system. Moreover, the physical properties of these blocks must have notable discrepancies to enhance the separation of the different phases. However, such properties often pose huge difficulties. For example, a high melting point block will increase the difficulty of synthesis and processing, while a low solubility block will raise the difficulty of characterization. For these reasons, only a few different tricrystalline polymers have been reported so far: PEO-*b*-PCL-*b*-PLLA[11,40–45], PE-*b*-PCL-*b*-PLLA[12], and PE-*b*-PEO-*b*-PCL[46].

The catalyst switch strategy is a convenient and attractive method to tackle the problem of synthesizing block copolymers with contradicting block chemistry[47]. In 2014, we proposed the first organic catalyst switch strategy to achieve a base-to-acid switch and afforded well-defined polyether-*b*-polyester type block copolymers in one pot[48]. Later, our group and Zhao and coworkers developed base-to-base[49], organic-to-metal[50], and biased Lewis acid/base switching strategies[51]. These strategies increase the variety of easily accessible block copolymers. At the same time, Williams and coworkers developed a switchable metal catalyst that can be used to synthesize polycarbonate-*b*-polyester in one pot[52–54]. The development of the catalyst switch strategy paves the way to achieving block copolymer with much higher complexity.

In 2019, we reported the first synthesis of a tetracrystalline tetrablock quarterpolymer, PE-*b*-PEO-*b*-PCL-*b*-PLLA[50]. Starting from PE-OH obtained by C1 polymerization, a phosphazene superbase $^t$BuP$_4$ was used to polymerize ethylene oxide (EO) at 80 °C. Diphenyl phosphate (DPP, to neutralize the base) and Sn(Oct)$_2$ were then added to perform an organic-to-metal catalyst switch and continue the polymerization of ε-caprolactone (CL) and L-lactide (LA). The tetracrystalline polymer was then obtained and its crystallization and self-assembly were studied[55].

Polyglycolide (PGA) is one of the most promising biobased biodegradable polymers that can be synthesized from both bio-sources and petroleum feedstock[56,57]. Its extremely high melting point ($T_m$ ~ 220 °C) among the polyester family makes it an ideal block in a multicrystalline polymer to study crystallization and phase interplay, as it can be distinguished easily from other blocks with much lower $T_m$. However, the synthesis of PGA-based polymer was challenging. The most common way to obtain PGA homopolymer is by the ROP in melt of glycolide at 150–230 °C, which is not considered as living polymerization, resulting in PGA with high dispersity, and is hard to produce PGA-based block copolymers[56,58]. There are only a few examples of PGA-based double crystalline copolymers in the literature[59–61], but no PGA-based tricrystalline triblock terpolymer reported so far. Recently, we developed a fluoroalcohol-assisted catalyst switch

strategy and reported the first living/controlled polymerization of glycolide at room temperature[62]. This strategy enables the synthesis of PGA-based multicrystalline block polymers.

Here, we combined the organic-to-metal switch with a fluoroalcohol-assisted catalyst switch strategy to perform the synthesis of a pentacrystalline pentablock quintopolymer, PE-*b*-PEO-*b*-PCL-*b*-PLLA-*b*-PGA. The molecular structure was successfully demonstrated by 1D and 2D liquid-state/solid-state NMR spectroscopy, high-temperature size exclusion chromatography (HT-SEC), Fourier transform infrared spectroscopy (FTIR), and thermal gravimetric analysis (TGA). The existence of five different crystallites was confirmed by wide-angle X-ray diffraction (WAXS), differential scanning calorimetry (DSC), and variable-temperature solid-state NMR spectroscopy.

## Results and discussion

The general synthetic route to achieve pentacrystalline pentablock quintopolymer PE-*b*-PEO-*b*-PCL-*b*-PLLA-*b*-PGA is shown in Table 1. The fluoroalcohol, 1,3-bis(2-hydroxyhexafluoroisopropyl)benzene (HFAB), was mixed with anhydrous toluene and GA, before being added to the reaction mixture to carry out the fluoroalcohol-assisted catalyst switch strategy. The NMR spectra used to determine monomer conversion as well as the number-average molecular weight ($M_{n,NMR}$) for the pentablock starting from PE-OH$_{1.5k}$ (pentablock-1a) are provided in Supplementary Figs. 1–3 and Supplementary Figs 8–12. The polymerization results of pentablock-1a are summarized in Table 1. As a comparison, a pentablock without adding HFAB was also synthesized by the same method (pentablock-1b, Supplementary Fig. 4). All intermediate polymerizations reach high monomer conversion (>99%), excluding the possibility of forming a gradient or blocky structure. However, without the fluoroalcohol-assisted catalyst switch by HFAB, the GA conversion remains low (58.3%) even after a prolonged time (36 h, Supplementary Fig. 4), probably because the PGA block is insoluble in toluene and immediately precipitates out from the solution. The solution became translucent ~20 minutes after injection of GA (Supplementary Fig. 5) and eventually turned into a gel-like solid (Supplementary Fig. 6). In contrast, the polymerization mixture of pentablock-1a (with HFAB) remained clear over 10 hours and turned translucent only at high monomer conversion (Supplementary Fig. 7). Obviously, HFAB plays an important role in the solubilization of the PGA block and in facilitating the living/controlled ROP of GA. Another pentablock starting from PE-OH$_{7k}$ was also synthesized to demonstrate the reproducibility of this method (pentablock-2, Supplementary Tab. 1). However, solution-based analyses ($^1$H NMR and SEC) could not be performed on pentablock-2 because the polymer is not soluble in any solvents/mix-solvents tested.

**Table 1 | Synthesis, molecular weight, and thermal properties of the pentablock quintopolymer (pentablock-1a) and the corresponding precursors[a]**

Pentacrystalline Pentablock Quintopolymer

| Entry | Sample | $t^{b}$ (h) | Conv. (%)[c] | $M_{n,NMR}$[d] (kg mol$^{-1}$) | | $M_{n,SEC}$[f] (kg mol$^{-1}$) | $T_m/T_c$[g] (°C) | | | | | |
|---|---|---|---|---|---|---|---|---|---|---|---|
| | | | | each block | total[e] | | PE | PEO | PCL | PLLA | PGA |
| 1 | PE-OH$_{1.5k}$[h] | 18 | >99 | 1.5 | 1.5 | 2.0 | 109.3/102.9 | | | | |
| 2 | PE-b-PEO-1 | 15 | >99 | 16.6 | 18.1 | 32.2 | 101.1/89.4 | 60.9/39.5 | | | |
| 3 | PE-b-PEO-b-PCL-1 | 45 | >99 | 15.7 | 33.8 | 45.4 | 101.6/87.8 | 51.5/24.9 | 55.5/36.0 | | |
| 4 | PE-b-PEO-b-PCL-b-PLLA-1 | 24 | >99 | 27.5 | 61.3 | 58.0 | 103.0/66.8 | 41.4/18.2 | 41.4/18.2 | 168.6/83.1 | |
| 5 | PE-b-PEO-b-PCL-b-PLLA-b-PGA-1a | 20 | 86.2 | 5.8 | 67.1 | 60.6 | 102.1/67.1 | 39.5/19.1 | 39.5/19.1 | 161.3/88.2 | 206.6/150.0 |

[a]The polymerization was performed using PE-OH as macroinitiator at 80 °C in toluene via $^{t}$BuP$_4$/DPP/Sn(Oct)$_2$ catalyst switch strategy. The targeting molecular weight for each block is 1.5 kg mol$^{-1}$, 15.0 kg mol$^{-1}$, 15.0 kg mol$^{-1}$, 30.0 kg mol$^{-1}$, and 7.0 kg mol$^{-1}$, respectively.

[b]Reaction time for the last block.

[c]Conversion of the last block, determined by $^1$H NMR at 60 °C (600 MHz, toluene-$d_8$), except for the PGA block, where a mixture of toluene-$d_8$/HFIP-$d_2$ (~7/3 v/v) was used.

[d]Determined by $^1$H NMR of the purified sample at 60 °C (600 MHz, toluene-$d_8$), except for the PGA block, where a mixture of toluene-$d_8$/HFIP-$d_2$ (~7/3 v/v) was used.

[e]Total $M_{n,NMR}$ at each stage.

[f]Determined by SEC (polystyrene standards) in THF at 50 °C.

[g]$T_m$ and $T_c$ were determined by DSC (10 °C min$^{-1}$ for heating and cooling, under N$_2$ atmosphere).

[h]Prepared by polyhomologation of sulfoxonium methylene in toluene at 80 °C.

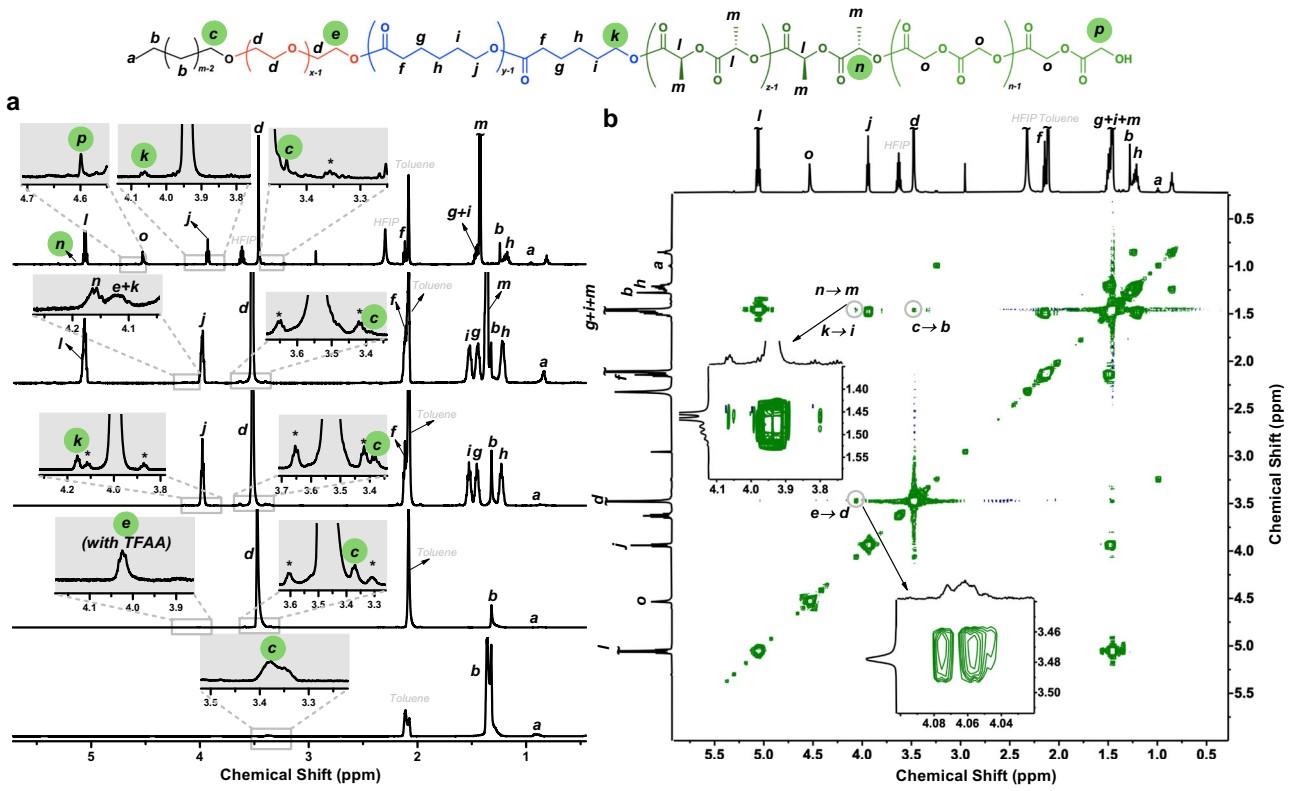

**Fig. 2 | Liquid-state NMR spectra of PE-*b*-PEO-*b*-PCL-*b*-PLLA-*b*-PGA-1a. a** 1D ¹H NMR spectra of pentablock-1a (Table 1, entries 1–5) and the corresponding precursors (Table 1, entries 1–5) are shown in Fig. 2a. The experiments were carried out on a 600 MHz spectrometer in toluene-*d*₈ at 60 °C (except for PE-OH, 80 °C, toluene-*d*₈, and the pentablock-1a, 60 °C, in a mixture of toluene-*d*₈/HFIP-*d*₂ (-7/3 v/v)). The green circles indicate the end group in each block.

The ¹H NMR spectra of the isolated pentablock-1a and the corresponding precursors (Table 1, entries 1–5) are shown in Fig. 2a. The -C*H₂*- signal (b) of the main chain was found at 1.47–1.18 ppm, while the -C*H₂*OH signal (c) of the chain-end was found at 3.42–3.32 ppm. The ratio of these two integrals was used to determine the $M_{n,NMR}$ of the PE precursor. The -C*H₂*CH₂O- signal (d) of the PEO main chain in PE-*b*-PEO was observed at 3.49 ppm, while the [-C*H₂*OH-TFAA] signal (e) was observed at 4.03 ppm. After the terpolymerization with CL, five new peaks corresponding to PCL merged. The peak at 4.00 ppm (j) was assigned to –C=O(CH₂)₅C*H₂*O-, and the integral was used to determine the $M_{n,NMR}$ of this block. The peak of -C = OC*H₂*(CH₂)₃O- appeared at 2.17–2.13 ppm (f). The remaining three methylene signals (-C*H₂*CH₂CH₂-, -CH₂C*H₂*CH₂-, -CH₂CH₂C*H₂*-) appeared at 1.49 ppm (i), 1.56 ppm (g), and 1.31–1.21 ppm (h), respectively. After the polymerization of LA, two peaks corresponding to -C=OC*H*CH₃- (5.09 ppm, l) and C=OCHC*H₃*- (1.40 ppm, m) appeared. In order to further confirm the successful incorporation of the PLLA block, TFAA was added to modify the end group. The appearance of a peak at −78.22 ppm on ¹⁹F NMR spectrum after the modification of the end group shows clearly the existence of the secondary hydroxyl group which belongs to the PLLA chain-end [-CH(CH₃)OH-TFA] (Supplementary Fig. 13). After the ROP of GA, the signal for methylene group -C=OC*H₂*O- was found at 4.53 ppm (o), confirming the successful synthesis of the pentablock quintopolymer. In addition, ¹⁹F NMR analysis of the trifluoroacetyl ester-derived pentablock (pentablock-TFA) revealed the existence of only the primary hydroxyl group (Supplementary Fig. 13, ¹⁹F signal at −76.16 ppm), indicating that there is no transesterification caused by the attack of PGA living chain-end to the PLLA block.

The correlation proton signal assignment is critical to confirm the connection of the five different blocks. For this reason, in addition to the 1D proton NMR, 2D NMR was performed for PE-*b*-PEO-*b*-PCL-*b*-PLLA-*b*-PGA-1a (pentablock-1a) to help assign the peaks for correlated protons. The correlation of the PE main chain with PE/PEO junction signal (c → b), the correlation of the PEO main chain with PEO/PCL junction signal (e → d), and the correlation of the PCL main chain with PCL/PLLA junction signal (k → i) were clearly observed on ¹H–¹H correlation spectroscopy (COSY, Fig. 2b). These assignments were also confirmed by total correlation spectroscopy (TOCSY, Supplementary Fig. 14), ¹H–¹³C heteronuclear single quantum coherence spectroscopy (HSQC, Supplementary Fig. 15), and ¹H–¹³C heteronuclear multiple bond correlation spectroscopy (HMBC, Supplementary Fig. 16). The ¹³C signals from carbonyl groups of the polyester (Supplementary Fig. 17) were also presented clearly by HMBC. Furthermore, the signals of -*C*=O, -(*C*)H, -*C*H₂-, and -*C*H₃ of PLLA and PGA (i, j, k, l, and m) appear as singlet peaks, corroborating that there is no lactyl-glycolyl or glycolyl-lactyl units formed due to transesterification from the attack of PGA living chain-end to the PLLA block. The signal at 176.2 ppm was assigned to PCL (4.0 ppm), the signal at 167.4 ppm showed a strong correlation with -C=OC*H₂*- (4.53 ppm, PGA), and the signal at 170.5 ppm was correlated to the -C=OC*H*(CH₃)O- signal (5.09 ppm, PLLA). The structures of the pentablock quintopolymers were also confirmed by solid-state ¹H magic angle spinning (MAS) NMR spectroscopy, 1D ¹H–¹³C cross-polarization (CP) MAS NMR spectroscopy (Supplementary Figs. 18–20 for pentablock-1a; Supplementary Figs. 21, 22 for pentablock-2), and 2D ¹H–¹³C short-distance heteronuclear correlation spectroscopy (HETCOR, Supplementary Fig. 18b, See Solid-state NMR analysis section in Supplementary Discussion) The successful synthesis of pentablock quintopolymer was also confirmed by the five characteristic peaks corresponding to each block on FTIR (Refer to Supplementary Discussion, FTIR section, and Supplementary Figs. 23–25).

The molecular weight and distribution were studied by SEC. HT-SEC was first used for the PE-OH₁.₅ₖ precursor, showing a narrow and

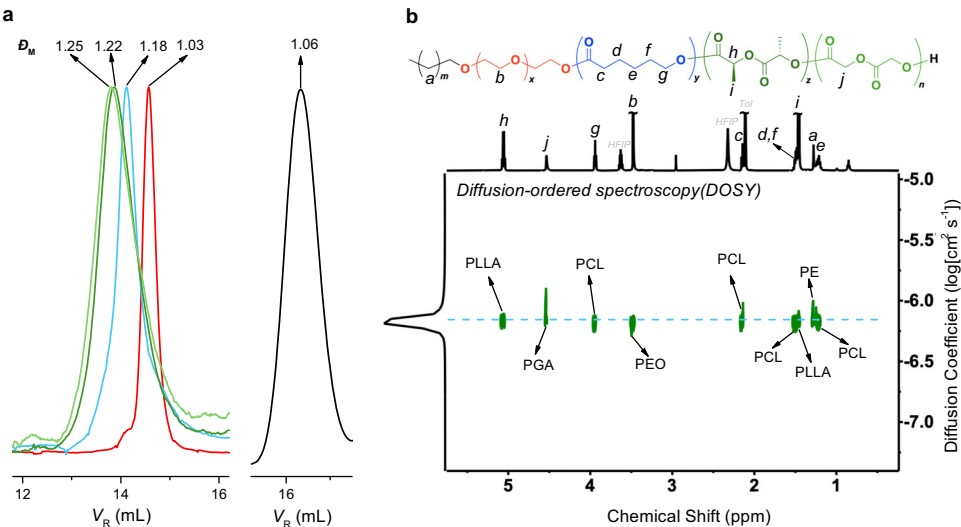

**Fig. 3 | SEC and DOSY spectrum of PE-*b*-PEO-*b*-PCL-*b*-PLLA-*b*-PGA-1a. a** SEC traces of PE-OH$_{1.5k}$ precursor (black), PE-*b*-PEO-1 (red), PE-*b*-PEO-*b*-PCL-1 (blue), PE-*b*-PEO-*b*-PCL-*b*-PLLA-1 (dark green), and pentablock-1a (light green). PE-OH$_{1.5k}$ trace was obtained on HT-SEC (TCB, 150 °C). The other traces were obtained on SEC with THF as eluent (50 °C). **b** Diffusion-ordered spectroscopy (DOSY) of pentablock-1a.

monomodal distribution ($Đ = 1.06$, Fig. 3a; $Đ = 1.15$ for PE-OH$_{7k}$, Supplementary Fig. 26) while SEC with THF as eluent was used at 50 °C for the rest of the samples[50]. All traces remain monomodal and exhibit low dispersity, which confirms that the five blocks are covalently linked within one macromolecule rather than a physical blend of five blocks.

To further demonstrate that the five crystalline blocks coexist in one molecule, diffusion-ordered spectroscopy (DOSY) was carried out at 60 °C (Fig. 3b, full spectrum in Supplementary Fig. 27). Only one diffusion coefficient ($6.55 \times 10^{-7}$ cm$^2$ s$^{-1}$) corresponding to five blocks was observed on the DOSY spectrum. As a comparison, the DOSY spectrum of the mixture of PE-*b*-PEO-1, PCL, PLLA, and PGA homopolymers (of similar molecular weight and mass ratio) shows distinct diffusion coefficients (Supplementary Fig. 28). This controlled experiment is consistent with the SEC and 2D NMR results and strongly confirms the successful synthesis of the pentablock quintopolymer.

The thermal properties of the multicrystalline multiblock samples were evaluated by TGA and DSC (Fig. 4 (pentablock-1a) and Supplementary Fig. 29 (pentablock-2)). The 5% weight loss temperature ($T_{d,5\%}$) was determined for the pentablock quintopolymers and all precursors (Supplementary Fig. 30 (pentablock-1a) and Supplementary Fig. 31 (pentablock-2)) before performing DSC. All samples studied here have $T_{d,5\%}$ higher than 250 °C except PE-*b*-PEO (~200 °C). Significant differences were observed in $T_{d,5\%}$ in pentablock-1a and pentablock-2 as well as for each individual block, which are mainly due to the different composition. Based on DSC, the $T_{m,PE}$ (Fig. 4a) was decreased from 109.3 °C to 101.1 °C from PE-OH$_{1.5k}$ (black) to PE-*b*-PEO-1 (red), this could be due to a confinement effect that deserves a separate study. The $T_{m,PE}$ then remained almost unchanged (101–103 °C) in tri-, tetra-, and pentablock polymers. The endothermic peaks of PCL and PEO merged due to the similar $T_m$. For this reason, Table 1 reports the same melting point for the two blocks for the cases where the endotherms overlap. The melting points of PEO/PCL kept decreasing from 60.9 °C to 39.5 °C as the different blocks were added, probably as a consequence of miscibility/dilution effects that will be studied in the following paper (i.e., the PLLA block is usually miscible with the PEO block, but the miscibility of the phases can be complicated when multiple blocks are present and should be studied carefully). The $T_{m,PLLA}$ was observed at 168.6 °C in the tetrablock quarterpolymer (dark green) and 161.3 °C in the pentablock quintopolymer (light green). Notably, a clear melting peak of PGA (206.6 °C) was observed in the pentablock quintopolymer (light green),

indicating the existence of five different crystalline phases in the solid state. Because of the overlap between the melting endotherms of PEO/PCL, the DSC studies can only clearly detect 4 endotherms at different temperatures. Nevertheless, applying a slow cooling/heating rate (1 °C min$^{-1}$) slightly separates the melting peaks of PEO and PCL (Supplementary Fig. 32), despite that, the melting peak of PGA cannot be observed probably due to thermal decomposition. Also, if the DSC results are combined with the WAXS and NMR data (see below), we can conclude that five different crystalline phases are present in the pentablock polymer.

The discussion about glass transition ($T_g$) is omitted in this paper, because the determination of $T_g$ (related exclusively to the amorphous regions of the sample) in multiphasic semi-crystalline materials is not easy (and beyond the scope of the present work), especially when the low $T_g$ values of PEO and PCL blocks would probably dominate if all five amorphous types of block chains form a weakly segregated or miscible phase. A fast chip calorimetry experiment is required to resolve the small changes in heat capacity (Cp) related to the $T_g$, where we will be able to quench the sample at 40.000 °C/s. In that case, the material will be rendered most probably 100% amorphous. Then, upon heating, we should be able to resolve the $T_g$ for the pentablock and all its precursors.

To the best of our knowledge, this is the first example of a pentablock quintopolymer with five different crystalline phases. The pentablock-2 sample shows a similar behavior except for a higher $T_m$ for the PE block (126.8 °C, Supplementary Fig. 29).

The crystallization behavior was studied for all samples after melting at 180 °C (225 °C for the pentablock) for 3 minutes (Fig. 4b). The $T_{c,PE}$ was suppressed from 102.9 °C for PE homopolymer (black) to 89.4 °C for PE-*b*-PEO-1 (red). It further decreased to 67.1 °C in the pentablock quintopolymer (light green). The $T_{c,PEO}$ and $T_{c,PCL}$ were observed separately in PE-*b*-PEO-*b*-PCL-1 (blue) to be 24.9 °C and 34.0 °C. The crystallization peaks of PEO and PCL merged into one broad peak in the tetrablock (dark green) and pentablock polymer (light green, 18.2 °C and 19.1 °C, respectively). Despite the low molecular weight of the PGA block (5.8 kg mol$^{-1}$), the crystallization peak for PGA (150.0 °C) was observed clearly in the DSC cooling curve. In our previous study, the crystallization of PLLA was very weak and required a slow cooling rate (1 °C min$^{-1}$) for the crystallization peak to be observed[50]. In this case, however, a strong peak of PLLA was observed at 88.2 °C, even with a fast cooling rate of 10 °C min$^{-1}$. This can be

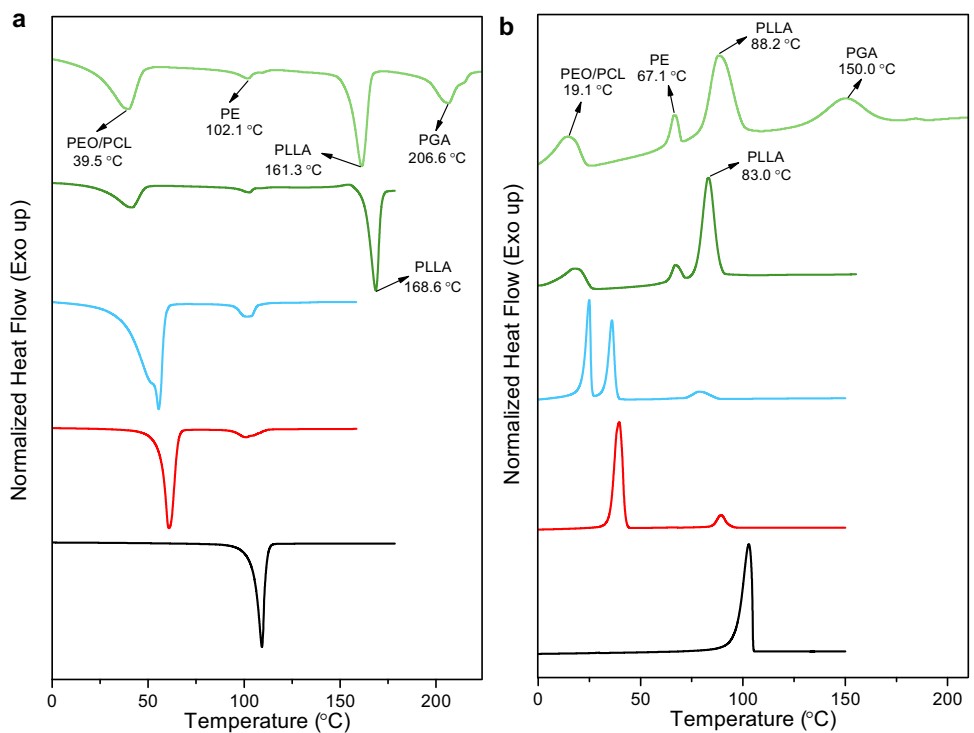

**Fig. 4 | Differential scanning calorimetry (DSC) scans of PE-*b*-PEO-*b*-PCL-*b*-PLLA-*b*-PGA-1a (pentablock-1a, light green) and all the precursors. a** Heating scan (10 °C min⁻¹). **b** Cooling scan (10 °C min⁻¹).

attributed to the higher PLLA relative content in the pentablock, and the nucleating action of the PGA crystals on the PLLA crystallization. This nucleating action is evidenced here by the increase in the crystallization temperature of the PLLA block when comparing the tetrablock and the pentablock polymers (see Fig. 4b).

In our previous work[55], we investigated the thermal properties of a PE₁₈-*b*-PEO₃₇-*b*-PCL₂₆-*b*-PLLA₁₉ tetrablock quarterpolymer. Several advanced characterization techniques were employed to identify with high precision the crystallization sequence of the blocks upon cooling from the melt: PLLA first, then PE, then PCL, and finally PEO. The corresponding reverse sequence was observed during subsequent heating. A similar behavior was observed on tetrablock-1 as shown in Fig. 4, except for the magnitude of the first-order thermal transitions (and their scan rate dependence), whose enthalpies were different because of the different compositions and molecular weights in the two tetrablocks. In the future, we will perform in situ WAXS/SAXS experiments at the synchrotron to determine precisely the sequential crystallization and melting of the pentablocks and their corresponding precursors.

To further prove the existence of five crystalline domains, mobility-dependent solid-state NMR magnetization transfer patterns were strategically employed to elucidate the structure based on mobility: mobile (i.e., with fast tumbling/rotation caused by amorphous/melted polymers) and rigid (crystalline) using scalar through-bond with ¹H–¹³C insensitive nuclei enhanced by polarization transfer (INEPT) technique and dipolar through-space ¹H–¹³C CP–MAS magnetization transfer techniques, respectively[63–65] (Fig. 5). The 1D variable-temperature (VT) ¹H–¹³C CP–MAS NMR experiments which give signals for only the immobile or crystalline domain were first carried out (Fig. 5a). At room temperature (RT), it is clear that all carbon signals from the CP spectrum are observed, indicating that all blocks are semi-crystalline. When the temperature increased to 60 °C, the peaks at 174.7 ppm (α, PCL), 71.5 ppm (d, PEO), 66.5 ppm (j, PCL), 34.3 ppm (f, PCL), 30.3 ppm (h, PCL) and 26.7 ppm (g + i, PCL) disappeared or significantly weakened. This means that the chains of PEO and became

mobile at this temperature, which is consistent with the melting point of PEO and PCL (39.5 °C) by DSC. At the same time, the peaks corresponding to PE (b, 33.8 ppm), PLLA (β, 171.5 ppm; l, 70.4 ppm; and m, 17.7 ppm), and PGA (γ, 169.0 ppm and o, 63.5 ppm) could still be observed on the spectrum. This means that at 60 °C, the PE, PLLA, and PGA chains are still rigid/semi-crystalline due to their higher melting points. When the temperature was increased to 120 °C, the peak corresponding to PE (b, 33.8 ppm) disappeared, proving the melting of the PE crystals at this temperature, which is consistent with the DSC results (102.1 °C).

In contrast to the VT ¹H–¹³C CP–MAS NMR spectra, only the flexible/mobile carbon signal can be observed on the VT ¹H–¹³C INEPT NMR spectrum (Fig. 5b). At RT, no signal is observed, implying that there are no mobile chains at RT. However, as the temperature increased to 60 °C, the peaks corresponding to PEO (d, 71.5 ppm) and PCL (j, 66.5 ppm; f, 34.3 ppm; h, 30.3 ppm; g, 26.7 ppm; and I, 26.7 ppm) appeared, indicating the melting of the PEO and PCL blocks. When the temperature further increased to 120 °C, a new peak appeared at 29.5 ppm. Interestingly, a very weak signal at 17.7 ppm corresponding to PLLA also appears at 120 °C, indicating that a small fraction of PLLA starts to melt at this temperature. This is probably because of the mobility transfer from the melted amorphous block to the semi-crystalline PLLA block due to mechanical spinning. A control experiment using low molecular weight PLLA (1.4 kg mol⁻¹) homopolymer shows no INEPT signal at 120 °C (Supplementary Fig. 33), excluding the possibility of forming oligomeric PLLA chains due to transesterification or other side reactions under such conditions.

The 2D ¹H–¹³C CP–MAS wide-line separation (WISE) solid-state NMR spectroscopy is another simple and unique method to probe the chain mobility and structure of the polymer[50,66,67]. The ¹³C signal shows the chemical structures where the line widths of the proton signals reflect the chain mobility. A typical line width of a semi-crystalline polymer or a rigid chain is over 50 kHz[68], whereas the line width for a flexible chain or an amorphous polymer is much sharper[67]. The WISE experiment was carried out at RT, 60 °C, and 120 °C for pentablock-1a

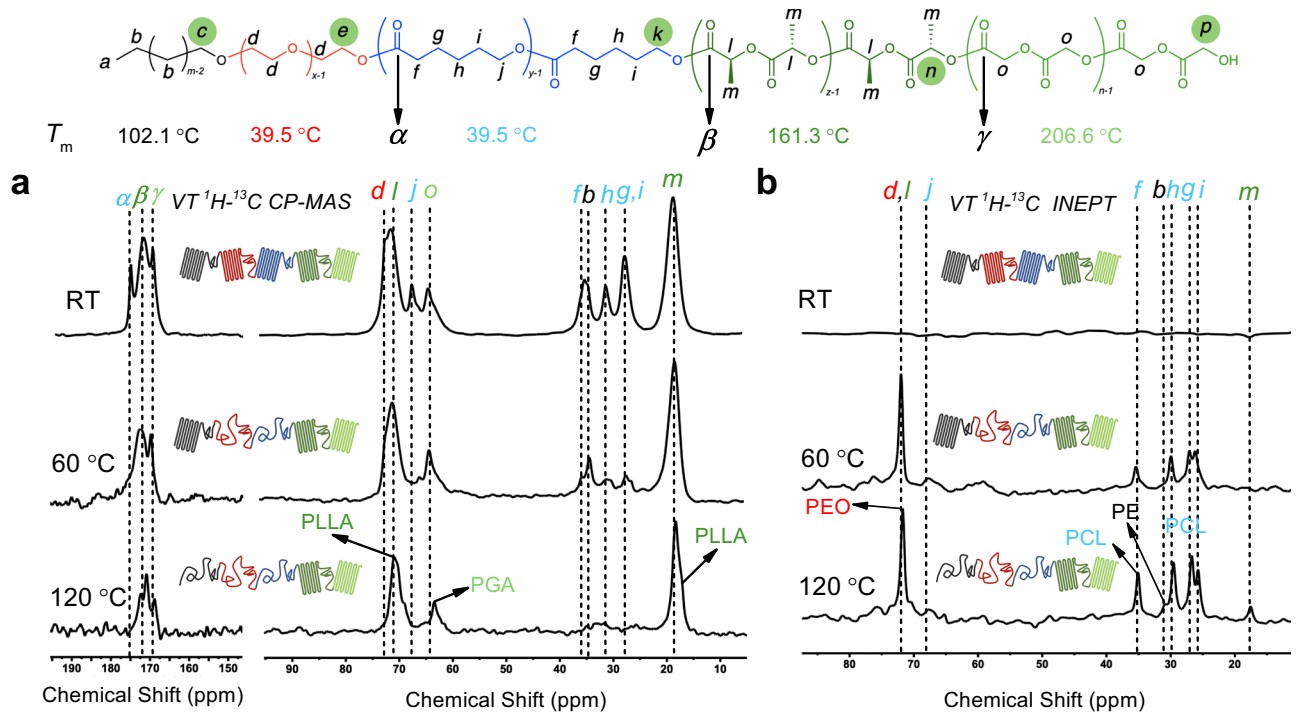

**Fig. 5 | Variable-temperature solid-state NMR spectroscopy. a** $^1$H–$^{13}$C CP-MAS NMR spectra and **b** $^1$H–$^{13}$C INEPT NMR spectra of PE-*b*-PEO-*b*-PCL-*b*-PLLA-*b*-PGA-1a (pentablock-1a) at different temperatures (RT, 60 °C and 120 °C). The green circles indicate the end group in each block. RT: room temperature.

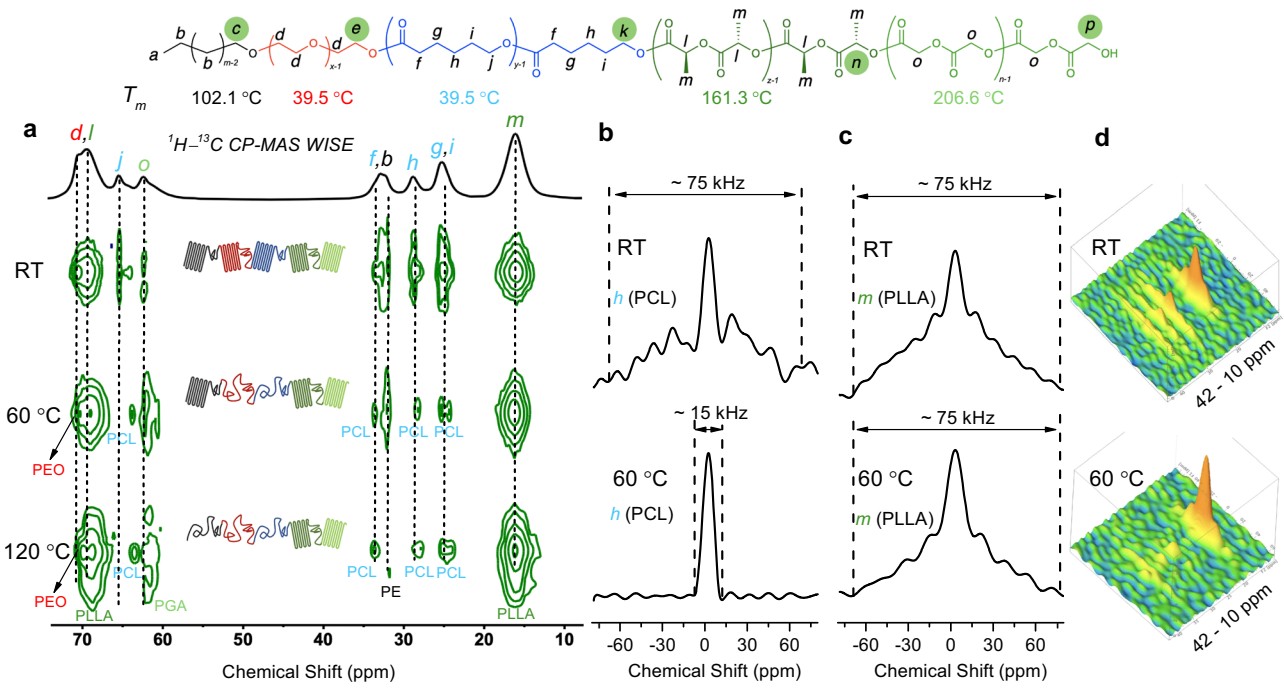

**Fig. 6 | Variable-temperature solid-state NMR spectroscopy. a** 2D $^1$H – $^{13}$C CP – MAS WISE spectra at RT, 60 °C, and 120 °C. **b** Slices extracted from the WISE spectrum at RT and 60 °C (30.3 ppm, PCL). **c** Slices extracted from the WISE spectrum at RT and 60 °C (17.7 ppm, PLLA). **d** 3D visualization of the WISE spectrum from 42–10 ppm. The green circles indicate the end group in each block. RT: room temperature.

(Fig. 6). The $^{13}$C signal of PE (-*C*H$_2$-, 33.8 ppm), PEO (-*C*H$_2$*C*H$_2$O-, 71.5 ppm), PCL (-*C*=O(CH$_2$)$_2$*C*H$_2$CH$_2$O-, 26.7 ppm), PLLA (-*C*=OCH(*C*H$_3$)-, 17.7 ppm), and PGA (-*C*=O*C*H$_2$O-, 63.5 ppm) were analyzed as representatives of the five blocks. All signals correspond to wide proton signals over 70 kHz, indicating low chain mobility of crystalized blocks at room temperature. When the temperature was increased to 60 °C,

the signal of PEO and PCL became much sharper (e.g., peak width ~15 kHz for PCL, Fig. 6b), indicating mobile polymeric chains. The other PE, PLLA, and PGA signals remained broad at this temperature (e.g., peak width ~75 kHz for PLLA, Fig. 6c). At 120 °C, the signal of PE became much sharper, where the peaks corresponding to PLLA and PGA were still broad and clear. The 3D images of the WISE spectra ranging from

42 to 10 ppm at RT and 60 °C clearly show the sharp signal of PCL above its melting point, where the PLLA and PE peaks are kept wide at 60 °C. These results further proved that all five blocks in the penta-block quintopolymer are semi-crystalline at room temperature and strengthened our conclusion.

WAXS was finally used to analyze whether all blocks of PE-*b*-PEO-*b*-PCL-*b*-PLLA-*b*-PGA and the corresponding precursors could be crystallized (Supplementary Fig. 34 for pentablock-1a, Supplementary Fig. 35 for pentablock-1b and Supplementary Fig. 36 for pentablock-2). The (110) and (200) characteristic reflections of the PE block were found at $2\theta = 21.7°$ and $23.8°$ in PE-*b*-PEO-*b*-PCL-*b*-PLLA-*b*-PGA-1a. A diffraction peak of PEO crystals was found at $2\theta = 19.4°$ [(120)] and $23.8°$ [(200)]. The WAXS reflections for PE and PCL overlap. The (010), (200/110), and (203) reflections of PLLA were clearly observed at $2\theta = 15.1°$, $17.0°$, and $19.4°$, respectively. Finally, the diffraction peaks of PGA were found at $2\theta = 22.5°$ [(110)] and $29.2°$ [(020)]. Despite its relatively low molecular weight, the strong diffraction peaks evidence a high degree of crystallinity for the PGA block. For the pentablock-2 sample, the WAXS diffractogram shows an even stronger diffraction peak at $2\theta = 22.5°$ (the (020) plane of PGA).

In summary, we have presented a fluoroalcohol-assisted catalyst switch strategy to incorporate a high melting point PGA block into a complex macromolecular architecture. Pentacrystalline pentablock quintopolymer PE-*b*-PEO-*b*-PCL-*b*-PLLA-*b*-PGA was successfully synthesized by combining polyhomologation (C1 polymerization), ROP, and the catalyst switch strategy. The use of fluoroalcohol as a co-solvent and co-catalyst prevented the crystallization of PGA during the reaction and produced PGA with a much higher conversion and significant melting/crystallization peak than that without fluoroalcohol. This method can be generalized for incorporating PGA block in any macromolecular architecture for PGA-based polymers.

The microstructure of the pentacrystalline pentablock quintopolymer was characterized by 1D/2D liquid-state NMR, 1D/2D solid-state NMR, SEC, FTIR, and TGA. The coexistence of five crystallites was confirmed by DSC, WAXS, variable temperature $^1$H–$^{13}$C CP-MAS NMR, variable temperature $^1$H–$^{13}$C INEPT NMR, and variable temperature $^1$H–$^{13}$C CP–MAS WISE NMR spectroscopy. The synthetic method reported here will open new horizons for the synthesis of complex multicrystalline macromolecules. The pentacrystalline pentablock quintopolymer will contribute to the advancement of polymer physics and the crystallization behavior of multiphasic soft matter.

## Methods

**PE-OH$_{1.5k}$** (Table 1, entry 1). 40.0 g of trimethylsulfoxonium iodide was added into a biphasic mixture of DCM (280 mL) and water (400 mL), followed by the addition of 61.5 g of tributylbenzylamino chloride. The mixture was kept in a dark place, for 24 hours. The aqueous phase was extracted and washed with DCM (70 mL × 3). After the removal of water on a rotary evaporator, a white needle-like crystal was obtained. The crystal was dissolved in methanol at 50 °C followed by cooling first to RT, and then to 0 °C in a refrigerator overnight. A needle-like crystal product of trimethylsufoxonium chloride (19.5 g, yield: 84%) with high purity (>99%) was obtained and confirmed by $^1$H and $^{13}$C NMR in deuterated dimethyl sulfoxide.

Then, 10.8 g (0.45 mol) of NaH was inserted into a two-necked flask connected to a reflux condenser and Ar/vacuum line. After three successive vacuum-air cycles, 180 mL of freshly distilled THF was injected. This was followed with the addition of 32.0 g (0.38 mol) of dry trimethylsulfoxonium chloride. The mixture was heated and refluxed at 80 °C until the gas ceased (5–6 h). The THF was removed under reduced pressure followed by the transfer of 150 mL of dry toluene into the flask. After stirring for 0.5 h, the resulting turbid solution was filtered through a dry Celite 545 column. A clear and transparent solution with a light yellow color was obtained. The concentration of ylide was determined by titration with a standard hydrochloric acid

aqueous solution (1.05 mmol mL$^{-1}$; yield 72%). The resulting ylide solution was transferred into a two-neck 1000 mL flask and kept under stirring for 30 min. Then, 1.90 mL of triethylborane (TEB, 1.90 mmol) was transferred into the flask to initiate the polymerization, and the solution was left for 30 min under stirring. After consumption of methylide, a tenfold excess (19.00 mmol, 1.90 g) of TAO·2H$_2$O was added to oxidize/hydrolyze the tris-organoborane at 80 °C to obtain PE-OH quantitatively. After 16 h, the reaction was stopped by cooling in an ice bath. The mixture was poured into cold methanol (300 mL), and the white solid was filtered, dried under vacuum, and characterized by HT-SEC and $^1$H NMR.

$^1$H NMR (600 MHz, toluene-*d*$_8$, 80 °C): δ/ppm = 3.42–3.32 (*m*, -C*H*$_2$-OH), 1.47–1.18 (*m*, -C*H*$_2$-), 0.91 (*ddd*, J = 18.7, 9.3, 4.9 Hz, -C*H*$_3$). $M_{n,NMR}$(PE-OH) = 1.5 kg mol$^{-1}$.

**PE-*b*-PEO-1** (Table 1, entry 2). 200 mg PE-OH precursor (0.229 mmol, 1.0 equiv.) was mixed with 12 mL dry toluene and 143 μL $^t$BuP$_4$ (0.1145 mmol, 0.5 equiv.) in a Schlenk tube inside a glovebox. The mixture was heated at 80 °C for 30 minutes before the addition of 2.76 mL ethylene oxide (55.17 mmol, 240.0 equiv., stored at −40 °C inside the glovebox). The mixture was then heated at 80 °C for 15 hours. 3 mL of the mixture was withdrawn and precipitated in -100 mL MeOH/Et$_2$O (2:8 v/v), centrifugated, and dried at room temperature overnight (684 mg, conv. > 99%). The remaining solution was used for the next block. $^1$H NMR (600 MHz, toluene-*d*$_8$, 80 °C, with one drop of trifluoroacetic acid): δ/ppm = 4.03 (*s*, PEO, [-CH$_2$C*H*$_2$-OH]) 3.49 (*s*, PEO, [-OC*H*$_2$C*H*$_2$O-]), 1.35 (*s*, PE, [-C*H*$_2$-]), 0.91 (*s*, PE, [-C*H*$_3$]). $M_{n,NMR}$(PEO) = 16.6 kg mol$^{-1}$.

**PE-*b*-PEO-*b*-PCL-1** (Table 1, entry 3). 25.8 mg DPP (0.103 mmol, 0.6 equiv.) was added to the PE-*b*-PEO-1 living solution (0.178 mmol). 1710 μL *ε*-caprolactone (15.46 mmol, 90.0 equiv.) and 20.9 mg Sn(Oct)$_2$ (0.052 mmol, 0.3 equiv.) were then added to the solution. Dry toluene was added to adjust the total volume to 20 mL before being heated at 80 °C for 45 hours. An aliquot was taken to determine the conversion (>99%). 2 mL solution was precipitated in cold MeOH/Et$_2$O (5:5 v/v), centrifugated, and dried at room temperature overnight (371.5 mg, >99%). The remaining solution was stored in the glovebox for the next block. $^1$H NMR (600 MHz, toluene-*d*$_8$, 80 °C): δ/ppm = 4.16 (*t*, PCL, [-COCH$_2$CH$_2$CH$_2$CH$_2$C*H*$_2$OH]), 4.00 (*t*, J = 6.7 Hz, PCL, [-COCH$_2$CH$_2$CH$_2$CH$_2$C*H*$_2$O-]), 3.54 (*s*, PEO, [-OC*H*$_2$C*H*$_2$O-]), 2.17 − 2.13 (*m*, PCL, [-COC*H*$_2$CH$_2$CH$_2$CH$_2$CH$_2$O-]), 1.56 (*p*, J = 7.9 Hz, PCL, [-COCH$_2$CH$_2$CH$_2$C*H*$_2$CH$_2$O-]), 1.49 (*p*, J = 7.0 Hz, PCL, [-COCH$_2$C*H*$_2$CH$_2$CH$_2$CH$_2$O-]), 1.36 (*s*, PE, [-C*H*$_2$-]), 1.31 − 1.21 (*m*, PCL, [-COCH$_2$CH$_2$C*H*$_2$CH$_2$CH$_2$O-]), 0.87 (*s*, PE, [-C*H*$_3$]). $M_{n,NMR}$(PCL) = 15.7 kg mol$^{-1}$.

**PE-*b*-PEO-*b*-PCL-*b*-PLLA-1** (Table 1, entry 4). 1039.5 mg L-lactide (7.21 mmol, 140.0 equiv.) was dissolved in 9 mL toluene before adding to 6 mL PE-*b*-PEO-*b*-PCL-1 living solution (0.052 mmol). The mixture was heated at 80 °C for 24 hours before an aliquot was withdrawn to determine the conversion (98.5%). 5 mL of the mixture was precipitated in cold MeOH/Et$_2$O (7:3 v/v), centrifugated, and dried at room temperature overnight (622.3 mg, 86.5%). The remaining solution was stored in the glovebox for the next block. $^1$H NMR (600 MHz, toluene-*d*$_8$, 80 °C): δ/ppm = 5.09 (*q*, J = 7.2 Hz, PLLA, [-OCOC*H*CH$_3$-]), 4.00 (*t*, J = 6.7 Hz, PCL, [-COCH$_2$CH$_2$CH$_2$CH$_2$C*H*$_2$O-]), 3.54 (*s*, PEO, [-OC*H*$_2$C*H*$_2$O-]), 2.15 (*t*, J = 7.4 Hz, PCL, [-COC*H*$_2$CH$_2$CH$_2$CH$_2$CH$_2$O-]), 1.56 (*p*, J = 7.4 Hz, PCL, [-COCH$_2$CH$_2$CH$_2$C*H*$_2$CH$_2$O-]), 1.49 (*p*, J = 7.0 Hz, PCL, [-COCH$_2$C*H*$_2$CH$_2$CH$_2$CH$_2$O-]), 1.40 (*d*, J = 7.0 Hz, PLLA, [-OCOCH*CH*$_3$-]), 1.36 (*s*, PE, [-C*H*$_2$-]), 1.30 − 1.20 (*m*, PCL, [-COCH$_2$CH$_2$C*H*$_2$CH$_2$CH$_2$O-]), 0.91 (*s*, PE, [-C*H*$_3$]). $M_{n,NMR}$(PLLA) = 27.5 kg mol$^{-1}$.

**PE-*b*-PEO-*b*-PCL-*b*-PLLA-*b*-PGA-1a** (Table 1, entry 5). 120 mg glycolide (1.03 mmol, 60.0 equiv.) was dissolved in 1 mL dry toluene and 4 mL HFAB before adding to 5 mL PE-*b*-PEO-*b*-PCL-*b*-PLLA-1 living solution (0.0172 mmol). The mixture was kept at 80 °C for 20 hours before an aliquot was withdrawn to determine the conversion (86.2%). The mixture was precipitated in cold MeOH/Et$_2$O

(7:3 v/v), centrifuged, and dried at room temperature overnight (771 mg, 91.8%). $^1$H NMR (600 MHz, toluene-$d_8$, 80 °C): δ/ppm = 5.07 (*q*, J = 7.0 Hz, PLLA, [-OCOC*H*(CH$_3$)-]), 4.48 (*s*, PGA, [-OCOC*H$_2$*-]), 3.95 (*t*, J = 6.6 Hz, PCL, [-COCH$_2$CH$_2$CH$_2$CH$_2$C*H$_2$*O-]), 3.47 (*s*, PEO, [-OC*H$_2$*C*H$_2$*O-]), 2.14 (*t*, J = 7.4 Hz, PCL, [-COC*H$_2$*CH$_2$CH$_2$CH$_2$CH$_2$O-]), 1.49 (*dt*, J = 18.1, 7.7 Hz, PCL, [-COCH$_2$CH$_2$CH$_2$C*H$_2$*CH$_2$O-] and PCL, [-COCH$_2$C*H$_2$*CH$_2$CH$_2$CH$_2$O-]), 1.44 (*d*, J = 7.1 Hz, PLLA, [-OCOCHC*H$_3$*]), 1.30 (*d*, J = 12.7 Hz, PE, [-C*H$_2$*-]), 1.26–1.16 (*m*, PCL, [-COCH$_2$CH$_2$C*H$_2$*CH$_2$CH$_2$O-]). $M_{n,NMR}$(PGA) = 5.8 kg mol$^{-1}$.

## Data availability

The data that support the findings of this paper are available in the main text or in the Supplementary Information. All other data are available from the corresponding author upon request.

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

## Acknowledgements

The research is supported by King Abdullah University of Science and Technology (KAUST). Our former member, Dr. Yunlong Hou was acknowledged for helping us with the polyethylene precursors. AJM gratefully acknowledges support from the Basque Government through Grant IT1309–19 and R + D + i project PID2020–113045GB-C21 funded by MCIN/AEI/10.13039/501100011033/.

## Author contributions

N.H., V.L., and P.Z conceived the idea and designed the experiments. P.Z. carried out the synthesis. P.Z and V.L performed the characterization. E.A. helped the solid-state NMR studies. A.J.M. helped the interpretation of the crystallization behavior of the pentablock. N.H. supervised this research. All authors discussed and co-wrote the manuscript.

## Competing interests

The authors declare no competing interest.
