## [Peer Review File · Nature Communications]

Catalyst Switch Strategy Enabled A Single Polymer with Five Different Crystalline PhasesReviewers' Comments:

Reviewer #1:

Remarks to the Author:

This manuscript details the synthesis and sequential polymerization of ethylene oxide (EO), ϵ -caprolactone (CL), lactide (LA) and glycolide (GA) using a hydroxyl-terminated polyethylene macroinitiator (PE-OH) to produce PE-b-PEO-b-PCL-b-PLLA-b-PGA as a pentablock quinquopolymer with all crystalline domains. This pentablock copolymer is thoroughly characterized via 1D and 2D NMR spectroscopy (both in solution and solid-state), SEC methods and the thermal properties by DSC and TGA, and by WAXS was also confirmed the pentacrystalline copolymer structure.

In contrast to previous studies, the incorporation of the PGA block can be only achieved by a recently published methodology (J. Am. Chem. Soc. 2023, 145, 27, 14756–14765) in which a fluoro-containing alcohol prevents crystallization of the PGA to preserve the livingness of the polymerization and assist the 'switch' catalysis. The sequential tetrapolymerization of PE-b-PEO-b-PCL-b-PLLA has also been already proven (Angew. Chem. Int. Ed. 2019, 58, 16267–16274). However, I do feel that this work is important and warrants publication in Nature Communications for the following reasons: 1) Multiblocks are very demanding to prepare and this work breaks new ground; 2) They apply commercial monomers used at scale in industry and make a sophisticated new composition/block structure – this will be generally of interest and could, furthermore, result in some new property profiles. 3) The work will inspire others to prepare multi-blocks and to explore their properties – it breaks new ground and is important. Overall, I do recommend publication of this work – it is important, interesting and addresses some long-standing challenges in the field of multiblock polymers. .

I have some questions/comments for the authors to consider

Comments on the manuscript and ESI

- In page 6, the authors claim "All polymerizations reach high monomer conversion (> 99%), excluding the possibility of forming a gradient or blocky structure". However, the polymerization of the last block PGA assisted by HFAB only reached ca. 86%. Perhaps re-consider the statement?
- Table 1: Reaction times in the scheme don't appear to match those listed in the table for each step of the polymerization.
- Table 1: The authors may consider editing the table caption to clarify that the M_n ,NMR is referring to only the specific block newly added for each row, and not the M_n ,NMR of the overall block polymer at each stage, e.g. 16.6 kg mol⁻¹ is the M_n ,NMR for just the PEO block, not the PE-b-PEO-1 overall. It's not immediately obvious from the table that this is the case and was only fully understood when referring to the annotated 1H NMR spectra in the ESI. Additionally, the authors could include an extra column for the total M_n ,NMR at each stage of the block polymer formation, as well as one for M_n ,SEC to see how they compare to one another.
- Table 1 and Table S1: The current formatting for the columns for T_m / T_c makes it difficult to read the values, particularly those for T_c as these have become split across lines.
- Figure 2b: why is the SNR in the vertical spectrum of the 1H-1H COSY so poor? In the horizontal spectrum the signal for b can be observed, however, in the vertical spectrum this signal is clearly absent. Do the authors have an explanation for this?
- In the 2D NMR spectra, it would be useful to present zoom areas, especially on correlation between blocks. It would provide more clarity for the reader.
- The authors explain that the observation of reduction of T_m is evidence for a miscibility/dilution effect (which was also observed previously in Angew. Chem. Int. Ed. 2019, 58, 16267–16274). Can the authors correlate the glass transition temperature for the same effect?
- Could the T_m / T_c of the single peak for the PEO/PCL block be resolved by slow cooling/heating rates in the DSC?
- ESI Page S9, Table S1: see above point regarding M_n ,NMR reporting.
- ESI Page S9, Table S1: Except for the starting PE-OH being of higher molar mass, were the same conditions (e.g. monomer loading) used for each stage to prepare pentablock-2 as for pentablock-1a? If so, why do the M_n ,NMR values for the PEO-2, PCL-2, and PGA-2 blocks (entries 2, 3, and 5) differ significantly compared to PEO-1, PCL-1, and PGA-1a? The conversion for the PGA-2 block is also

missing from this table; its omission is not reflected in the table caption. The authors should also present the NMR spectroscopy evidence for the formation of this pentablock copolymer.

- ESI Page S22, Figure S29/S30: Could the authors provide an explanation for why the T_{d,5%} for pentablock-1a is so low at around 161 °C, and why this is so different compared to the value for pentablock-2? Further, why are the decomposition profiles for the diblock sample so different between sample 1 and sample 2?
- ESI Page S22, Figure S30: What is the reason for the abrupt step decrease in weight observed for PE-OH at approx. 430-440 °C?

Reviewer #2:

Remarks to the Author:

In this paper, Zhang et al. report the synthesis of the first pentablock quaterpolymer (PE-b-PEO-b-PCL-b-PLLA-b-PGA) presenting five different crystalline domains. These results are building up from previous work from the same group on the preparation of a tetracrystalline tetrablock quaterpolymer (PE-b-PEO-b-PCL-b-PLLA; *Angew. Chem. Int. Ed.* 2019, 58(45): 16267-16274). Using the similar elegant strategy combining polyhomologation, organo-catalyze ROP and organic-to-metal "catalyst switch", they successfully synthesized the initial tetrablock quaterpolymer which was then converted into the targeted pentablock quaterpolymer using a fluoroalcohol-assisted catalyst switch. They used a range of solution and solid state 1D and 2D NMR spectroscopy techniques, WAXS and thermal analysis to demonstrate the successful synthesis of the pentablock quaterpolymer and the existence of the five distinct crystalline domains. This manuscript has substantial strengths but perhaps needs to be strengthened prior to publication to Nature Communication.

1) The title is misleading. The data presented show only four different melting and crystallizing temperatures since two of these transition temperatures are the same (for PEO and PCL). The authors should change the title accordingly.

2) The authors prepared two pentablocks, one using the fluoroalcohol-assisted catalyst switch and one without. Without the presence of 1,3-bis(2-hydroxyhexafluoro-isopropyl)benzene (HFAB) they observed the formation of a gel despite low conversion of GA. In contrast, when using HFAB, they have a higher conversion of GA and no gel formation. The authors should comment on: Why are these differences observed? Is it only a solubility issue? Also, the authors should demonstrate that there is no evidence of trans-esterification happening for PE-b-PEO-b-PCL-b-PLLA-b-PGA-2. Another question that they should address is: Could it be that the apparent lower conversion of GA is due to the lack of solubility of PE-b-PEO-b-PCL-b-PLLA-b-PGA-2? To remove any doubts, the authors should therefore include a quantitative ¹H NMR spectrum to calculate the conversion of GA. The authors should also include all the SEC traces for the synthesis of PE-b-PEO-b-PCL-b-PLLA-b-PGA-2 as well as all the solution states characterization for this polymer.

3) Also related to my first point, one of the biggest claims in the paper is that the fluoroalcohol-assisted catalyst switch is key for the synthesis of these pentablocks. However, I fail to see the advantage of using this method because in both cases, it looks like a pentablock with similar properties can be isolated. The authors should make a clear comparison between PE-b-PEO-b-PCL-b-PLLA-b-PGA-1a and PE-b-PEO-b-PCL-b-PLLA-b-PGA-2 to thoroughly explain why this strategy is essential.

4) A range of solution state 1D and 2D NMR spectroscopy techniques are used to characterize PE-b-PEO-b-PCL-b-PLLA-b-PGA-1a. The data presented by the authors are mostly convincing. However, there are four points that the author need to address:

a. First, the COSY spectrum in Fig. 2b is of overall low quality. The authors should process the spectrum via COSY symmetrisation and an arbitrary T1 noise reduction algorithm. The correlation of

n→m can only be surmised, since the correlating symmetric signal is buried in the T1 noise streak of g+i+m. Adding to that, the authors should also improve the quality of the f2 trace, since the S/N ratio is high. There is also a visible cutoff on the f2 trace scale at approximately 0.5 ppm.

b. Second, in the ¹H-¹³C HMBC spectrum of PE-b-PEO-b-PCL-b-PLLA-b-PGA-1a (Fig. S15), one can see a cross peak at around 163 ppm aligned with methine peak of PLLA. Could the author comment on it? If it is a ghost peak, could the author comment on the risk that the peak they assigned as c→d, of similar relative intensity, is not also a ghost peak?

c. Third, the authors should describe in more details on the processing of the DOSY data. I refer them to this following paper: <https://doi.org/10.1002/macp.201900255> which show that a mixture of miscible homopolymers and block copolymers could lead to a single diffusion coefficient, if the data is not treated properly. They should ensure that their data treatment has been done properly.

d. Fourth, the authors should do a quantitative analysis of the end-group conversion by using a phosphorous-containing reagent such as 2-chloro-4,4,5,5-tetramethyl dioxaphospholane to monitor the change in the chemical shift of the phosphorylated hydroxyl end group as a function of the growing of the next block. This would rule out any possibility of trans-esterification and the formation of other multiblock copolymers.

5) Related to the possibility of trans-esterification, the author mention in the discussion of the VT ¹H-¹³C CP-MAS NMR spectra that "a very weak signal at 17.7 ppm corresponding to PLLA also appears at 120 °C, indicating that a small fraction of PLLA starts to melt at this temperature". They postulate that this could be do to "the mobility transfer from the melted amorphous block to the semi-crystalline PLLA block due to mechanical spinning". The authors should demonstrate that this is not due to transesterification issues which would have shorten the PLLA chain in some of the polymer and would give similar results.

6) The TGA results show significant change in the T_{5%} for PE-b-PEO-b-PCL-b-PLLA-b-PGA-1a and PE-b-PEO-b-PCL-b-PLLA-b-PGA-2 not only for the final polymers but also for each of the individual blocks. The authors should comment on these differences.

Minor Comments

7) The temperatures in the tables are difficult to read and the conversion in entry 5 in table S1 is missing even though the authors report a conversion in the text.

8) In the introduction, the authors wrote: "Recently, we developed a "fluoroalcohol-assisted catalyst switch" strategy (P.Z., V.L., and N.H.) and reported the first living/controlled polymerization of glycolide at room temperature." A reference is missing here. The authors should also remove or explain what the abbreviations in the brackets mean.

9) When the authors are discussing about the thermal properties of the different blocks, they should relate to their previous study and comment if similar (or different) behavior are observed.

RESPONSES TO THE REVIEWERS' COMMENTS

Our point-by-point responses to the reviewers' comments are given below in blue.

The additions/changes in the MS and SI are given in red.

Reviewer #1 (Remarks to the Author):

This manuscript details the synthesis and sequential polymerization of ethylene oxide (EO), ϵ -caprolactone (CL), lactide (LA) and glycolide (GA) using a hydroxyl-terminated polyethylene macroinitiator (PE-OH) to produce PE-b-PEO-b-PCL-b-PLLA-b-PGA as a pentablock quinterpolymer with all crystalline domains. This pentablock copolymer is thoroughly characterized via 1D and 2D NMR spectroscopy (both in solution and solid-state), SEC methods and the thermal properties by DSC and TGA, and by WAXS was also confirmed the pentacrystalline copolymer structure.

In contrast to previous studies, the incorporation of the PGA block can be only achieved by a recently published methodology (J. Am. Chem. Soc. 2023, 145, 27, 14756–14765) in which a fluoro-containing alcohol prevents crystallization of the PGA to preserve the livingness of the polymerization and assist the 'switch' catalysis. The sequential tetrapolymerization of PE-b-PEO-b-PCL-b-PLLA has also been already proven (Angew. Chem. Int. Ed. 2019, 58, 16267–16274). However, I do feel that this work is important and warrants publication in Nature Communications for the following reasons: 1) Multiblocks are very demanding to prepare and this work breaks new ground; 2) They apply commercial monomers used at scale in industry and make a sophisticated new composition/block structure – this will be generally of interest and could, furthermore, result in some new property profiles. 3) The work will inspire others to prepare multiblocks and to explore their properties – it breaks new ground and is important. Overall, I do recommend publication of this work – it is important, interesting and addresses some long-standing challenges in the field of multiblock polymers.

Answer: Thanks for the reviewer's positive comments.

I have some questions/comments for the authors to consider

Comments on the manuscript and ESI

- In page 6, the authors claim “All polymerizations reach high monomer conversion (> 99%), excluding the possibility of forming a gradient or blocky structure”. However, the polymerization of the last block PGA assisted by HFAB only reached ca. 86%. Perhaps re-consider the statement?

Answer: We intentionally let the polymerization for each intermediate block to reach almost quantitative monomer conversions in order to have a pure block structure. We added the word “intermediate” to make a precise statement.

Addition to the MS: “All intermediate polymerizations reach high monomer conversion (>99 %), ...”

- Table 1: Reaction times in the scheme don't appear to match those listed in the table for each step of the polymerization.

Answer: We updated the scheme to match the times listed in the table.

Correction to the reaction Scheme on Table 1:

- Table 1: The authors may consider editing the table caption to clarify that the M_n , NMR is referring to only the specific block newly added for each row, and not the M_n , NMR of

the overall block polymer at each stage, e.g. 16.6 kg mol⁻¹ is the Mn,NMR for just the PEO block, not the PE-*b*-PEO-1 overall. It's not immediately obvious from the table that this is the case and was only fully understood when referring to the annotated 1H NMR spectra in the ESI. Additionally, the authors could include an extra column for the total Mn,NMR at each stage of the block polymer formation, as well as one for Mn,SEC to see how they compare to one another.

Answer: We revised Table 1 accordingly. Mn, SEC information is updated accordingly as well. Due to the limited space, the dispersities are not shown here, but they are presented in Fig. 3. Please see the table in the next page.

• Table 1 and Table S1: The current formatting for the columns for Tm/Tc makes it difficult to read the values, particularly those for Tc as these have become split across lines.

Answer: Thanks for the suggestions. We revised the tables (Table 1 and Table S1) for better clarity. Please see below.

Corrections made to the following Tables:

Table 1. Synthesis, molecular weight, and thermal properties of the pentablock quinquopolymer (pentablock-1a) and the corresponding precursors.^[a]

Entry	Sample	t ^[c] (h)	Conv. (%) ^[d]	$M_{n,NMR}$ ^[e] (kg mol ⁻¹)		$M_{n,SEC}$ ^[g] (kg mol ⁻¹)	T_m/T_c ^[h] (°C)					
				each block	total ^[f]		PE	PEO	PCL	PLLA	PGA	
1	PE-OH _{1.5k} ^[b]	18	>99	1.5	1.5	2.0	109.3/ 102.9					
2	PE- b -PEO-1	15	>99	16.6	18.1	32.2	101.1/ 89.4	60.9/ 39.5				
3	PE- b -PEO- b -PCL-1	45	>99	15.7	33.8	45.4	101.6/ 87.8	51.5/ 24.9	55.5/ 36.0			
4	PE- b -PEO- b -PCL- b -PLLA-1	24	>99	27.5	61.3	58.0	103.0/ 66.8	41.4/ 18.2	41.4/ 18.2	168.6/ 83.1		
5	PE- b -PEO- b -PCL- b -PLLA- b -PGA-1a	20	86.2	5.8	67.1	60.6	102.1/ 67.1	39.5/ 19.1	39.5/ 19.1	161.3/ 88.2	206.6/ 150.0	

[a] The polymerization was performed using PE-OH as macroinitiator at 80 °C in toluene via ^tBuP₄/DPP/Sn(Oct)₂ “catalyst switch” strategy. [b] Prepared by polyhomologation of sulfoxonium

methylene in toluene at 80 °C. [c] Reaction time for the last block. [d] Conversion of the last block, determined by ¹H NMR at 60 °C (600 MHz, toluene-*d*₈), except for the PGA block, where a mixture of toluene-*d*₈/HFIP-*d*₂ (~7/3 v/v) was used. [e] Determined by ¹H NMR of the purified sample at 60 °C (600 MHz, toluene-*d*₈), except for the PGA block, where a mixture of toluene-*d*₈/HFIP-*d*₂ (~7/3 v/v) was used. [f] Total *M*_{n,NMR} at each stage. [g] Determined by SEC (polystyrene standards) in THF at 50 °C. [h] *T*_m and *T*_c were determined by DSC (10 °C min⁻¹ for heating and cooling, under N₂ atmosphere).

Table S1. Molecular weight and thermal information of the pentablock quinterpolymer (pentablock-2) and corresponding precursors.^[a]

Entry	Sample	t ^[c] (h)	Conv. (%) ^[d]	M _{n,NMR} ^[e] (kg mol ⁻¹)		T _m / T _c ^[g] (°C)					
				each block	total ^[f]	PE	PEO	PCL	PLLA	PGA	
1	PE-OH _{7k} ^[b]	18	>99	7.0	7.0	131.8/ 118.7					
2	PE- b -PEO- 2	15	>99	10.6	17.6	131.2/ 118.6	61.9/ 41.8				
3	PE- b -PEO- b -PCL-2	45	>99	11.4	29.0	131.1/ 117.8	56.2/ 29.6	56.2/ 44.0			
4	PE- b -PEO- b -PCL- b - PLLA-2	24	>99	25.9	54.9	128.8/ 96.6	39.9/ 36.4	39.9/ 36.4	155.7/ 114.3		
5	PE- b -PEO- b -PCL- b - PLLA- b - PGA-2	20	- ^[h]	- ^[h]	- ^[h]	126.7/ 90.5	41.8/ 42.0	41.8/ 42.0	147.6/ 119.1	209.0/ 144.3	

[a] Polymerization was performed with the PE-OH macroinitiator at 80 °C in toluene via ^tBuP₄/DPP/Sn(Oct)₂ catalyst switch strategy. The targeting molecular weight for each block is 7 kg mol⁻¹, 10 kg mol⁻¹, 10 kg mol⁻¹, 25 kg mol⁻¹, and 15 kg mol⁻¹, respectively. [b] Prepared by polyhomologation of sulfoxonium methylene in toluene at 80 °C. [c] Reaction time for the last block. [d] Conversion of the last block, determined by ¹H NMR at 60 °C (600 MHz, toluene-*d*₈), except for the PGA block, where a mixture of toluene-*d*₈/HFIP-*d*₂ (~7/3 v/v) was used. [e] Determined by ¹H NMR of the purified sample at 60 °C (600 MHz, toluene-*d*₈), except for the PE-OH_{7k} precursor, where a higher temperature of 90 °C was used. [f] Total *M*_{n,NMR} at each stage. [g] Determined by DSC (10 °C min⁻¹ for heating and cooling, under N₂ atmosphere). [h] The sample is not fully soluble, even in the mixture of HFAB and toluene at high temperatures.

• Figure 2b: why is the SNR in the vertical spectrum of the 1H-1H COSY so poor? In the horizontal spectrum the signal for b can be observed, however, in the vertical spectrum this signal is clearly absent. Do the authors have an explanation for this?

Answer: The COSY experiment was conducted at high temperatures to dissolve the PE block. It is well-known that at high temperatures, the ¹H signal tends to be broad and produce more noise. We reprocessed the COSY signal carefully and used an external

projection from 1D ^1H spectrum (performed just before the COSY experiment) for f1 and f2 dimensions.

We improved the quality of b. Modified Figure is given below

Fig. 2 | Liquid-state NMR spectra of PE-*b*-PEO-*b*-PCL-*b*-PLLA-*b*-PGA-1a. a. 1D ^1H NMR spectra of pentablock-1a (Table 1, entries 1-5) and the corresponding precursors. **b.** ^1H - ^1H COSY spectrum of pentablock-1a (Table 1, entries 5). The experiments were carried out on a 600 MHz spectrometer in toluene- d_8 at 60 °C (except for PE-OH, 80 °C, toluene- d_8 , and the pentablock-1a, 60 °C, in a mixture of toluene- d_8 /HFIP- d_2 (~7/3 v/v)).

- In the 2D NMR spectra, it would be useful to present zoom areas, especially on correlation between blocks. It would provide more clarity for the reader.

Answer: Thanks for the comment. We have added the zoom areas on the correlation between blocks.

- The authors explain that the observation of reduction of T_m is evidence for a miscibility/dilution effect (which was also observed previously in *Angew. Chem. Int. Ed.* 2019, 58, 16267–16274). Can the authors correlate the glass transition temperature for the same effect?

Answer: This is an interesting question, and the answer is maybe yes, but determining glass transition temperatures (T_g , related exclusively to the amorphous regions of the sample) in multiphasic semi-crystalline materials is not easy, especially when the low T_g values of PEO and PCL blocks would probably dominate if all five amorphous types of block chains form a weakly segregated or miscible phase. Usually, the signal in the DSC for these types of systems is not good enough to resolve the small changes in heat capacity (C_p) related to the T_g . However, we have plans to perform fast chip calorimetry in the near future, where we will be able to quench the sample at 40.000 °C/s, and in that case, the material will be rendered most probably 100% amorphous. Then, upon heating, we should be able to resolve the T_g for the pentablock and all its precursors. However, these experiments are beyond the scope of the present work.

Addition to the MS: The discussion about glass transition (T_g) is omitted in this paper, because the determination of T_g (related exclusively to the amorphous regions of the sample) in multiphasic semi-crystalline materials is not easy (and beyond the scope of the present work), especially when the low T_g values of PEO and PCL blocks would dominate if all five amorphous types of block chains forming a weakly segregated or miscible phase. A fast chip calorimetry experiment is required to resolve the small changes in heat capacity (C_p) related to the T_g , where we will be able to quench the sample at 40.000 °C/s. In that case, the material will be rendered most probably 100% amorphous. Then, upon heating, we should be able to resolve the T_g for the pentablock and all its precursors.

- Could the T_m/T_c of the single peak for the PEO/PCL block be resolved by slow cooling/heating rates in the DSC?

Answer: Normally, slower cooling and heating rates would only slightly separate the overlap melting peaks of PEO/PCL by revealing a bimodal signal but not completely

separating them. For this reason, we have run new DSC experiments at a rate of 1 °C/min during cooling from the melt and subsequent heating (Fig. S32). Indeed, the first melting peak at around 50 °C now has a bimodal character with a high-temperature shoulder that could be attributed to the sequential melting of PEO and PCL. This would need confirmation by in situ WAXS studies at the synchrotron, something we plan to perform in the future.

We added to SI the Fig. S32 corresponding to additional experiments.

Fig. S32. DSC curve of PE-*b*-PEO-*b*-PCL-*b*-PLLA-*b*-PGA-1a. The heating rate was 1 °C min⁻¹.

Addition to the MS:

Nevertheless, applying a slow cooling/heating rate (1 °C min⁻¹) slightly separates the melting peaks of PEO and PCL (**Fig. S32**), despite that, the melting peak of PGA cannot be observed probably due to thermal decomposition.

• ESI Page S9, Table S1: see above point regarding Mn,NMR reporting.

Answer: We revised Table S1 according to the reviewer's suggestion. Please see the revised Table S1 above.

• ESI Page S9, Table S1: Except for the starting PE-OH being of higher molar mass, were the same conditions (e.g. monomer loading) used for each stage to prepare pentablock-2 as for pentablock-1a? If so, why do the M_n , NMR values for the PEO-2, PCL-2, and PGA-2 blocks (entries 2, 3, and 5) differ significantly compared to PEO-1, PCL-1, and PGA-1a?

Answer: The targeting molecular weight is different for pentablock-1 and pentablock-2. The targeted molecular weights are now listed in the footnotes of Table 1 and Table S1. We targeted higher molecular weight for PEO, PCL, and PLLA in pentablock-1a to increase the overall solubility of the pentablock-1a for better characterization (e.g. SEC and various NMR).

Addition to the Tables: (pentablock-1a) The targeting molecular weight for each block is 1.5 kg mol^{-1} , 15 kg mol^{-1} , 15 kg mol^{-1} , 30 kg mol^{-1} , and 7 kg mol^{-1} , respectively. (pentablock-2) The targeting molecular weight for each block is 7 kg mol^{-1} , 10 kg mol^{-1} , 10 kg mol^{-1} , 25 kg mol^{-1} , and 15 kg mol^{-1} , respectively.

The conversion for the PGA-2 block is also missing from this table; its omission is not reflected in the table caption. The authors should also present the NMR spectroscopy evidence for the formation of this pentablock copolymer.

Answer: The omission of the conversion of the PGA block in pentablock-2 is due to the insoluble nature of the last block. We tried many times to dissolve pentablock-2 but it was not fully soluble. We added the following sentence on the footnote of Table S1.

Addition to the SI: [h] The sample is not fully soluble, even in the mixture of HFAB and toluene at high temperatures.

• ESI Page S22, Figure S29/S30: Could the authors provide an explanation for why the $T_{d,5\%}$ for pentablock-1a is so low at around $161 \text{ }^\circ\text{C}$, and why this is so different compared to the value for pentablock-2?

Answer: We have redone the TGA for pentablock-1. The $T_{d,5\%}$ is found around $199.9 \text{ }^\circ\text{C}$. The relatively low $T_{d,5\%}$ might be due to the residual catalyst accelerating the decomposition of the polymer.

New Figures S30 and S31 corresponding to repeated experiments:

Fig. S30. TGA thermograms of the PE-*b*-PEO-*b*-PCL-*b*-PLLA-*b*-PGA-1a (pentablock-1a, Table 1, entry 5) and all corresponding precursors (heating rate: 10 °C min⁻¹).

Fig. S31. TGA thermograms of the PE-*b*-PEO-*b*-PCL-*b*-PLLA-*b*-PGA-2 (pentablock-2, Table S1, entry 5) and all corresponding precursors (heating rate: 10 °C min⁻¹).

Reviewer #2 (Remarks to the Author):

In this paper, Zhang et al. report the synthesis of the first pentablock quintopolymer (PE-b-PEO-b-PCL-b-PLLA-b-PGA) presenting five different crystalline domains. These results are building up from previous work from the same group on the preparation of a tetracrystalline tetrablock quaterpolymer (PE-b-PEO-b-PCL-b-PLLA; *Angew. Chem. Int. Ed.* 2019, 58(45): 16267-16274). Using the similar elegant strategy combining polyhomologation, organo-catalyze ROP and organic-to-metal “catalyst switch”, they successfully synthesized the initial tetrablock quaterpolymer which was then converted into the targeted pentablock quintopolymer using a fluoroalcohol-assisted catalyst switch. They used a range of solution and solid state 1D and 2D NMR spectroscopy techniques, WAXS and thermal analysis to demonstrate the successful synthesis of the pentablock quintopolymer and the existence of the five distinct crystalline domains. This manuscript has substantial strengths but perhaps needs to be strengthened prior to publication to Nature Communication.

1) The title is misleading. The data presented show only four different melting and crystallizing temperatures since two of these transition temperatures are the same (for PEO and PCL). The authors should change the title accordingly.

Answer: The melting points of the PEO and PCL blocks overlap, but as observed above in the responses to reviewer 1, they can be separated by employing slower cooling and heating rates. Nevertheless, the separation is not complete. Hence, we decided to change the title to:

“Catalyst Switch” Strategy Enabled Pentacrystalline Pentablock Quintopolymer: A Single Polymer with Five Different Crystalline Phases”

2) The authors prepared two pentablocks, one using the fluoroalcohol-assisted catalyst switch and one without. Without the presence of 1,3-bis(2-hydroxyhexafluoroisopropyl)benzene (HFAB) they observed the formation of a gel despite low conversion

of GA. In contrast, when using HFAB, they have a higher conversion of GA and no gel formation. The authors should comment on: Why are these differences observed? Is it only a solubility issue?

Answer: The differences observed between PE-*b*-PEO-*b*-PCL-*b*-PLLA-*b*-PGA-1a (pentablock-1a) and PE-*b*-PEO-*b*-PCL-*b*-PLLA-*b*-PGA-1b (pentablock-1b) is mainly due to solubility issue. Pentablock-1b was synthesized in toluene at 80 °C. When the PGA block is formed, even at 56% GA conversion, the formed PGA block is insoluble and immediately precipitates out from the solution, resulting in a gel-like polymer. The solidified mixture cannot be melted even at higher temperatures (up to 130 °C). In addition, HFAB can also facilitate the living/controlled ROP of glycolide, which was described in detail in our recent paper (P. Zhang, V. Ladelta, and N. Hadjichristidis *J. Am. Chem. Soc.* 2023, 145, 27, 14756–14765).

Addition to the MS: However, without the “fluoroalcohol-assisted catalyst switch” by HFAB, the GA conversion remains incomplete (58.3%) even after a prolonged time (36 h, **Fig. S4**), **probably because PGA block is insoluble in toluene and immediately precipitates out from the solution.** The solution became translucent ~20 minutes after injection of GA (**Fig. S5**) and eventually a gel-like solid (**Fig. S6**).

Also, the authors should demonstrate that there is no evidence of trans-esterification happening for PE-*b*-PEO-*b*-PCL-*b*-PLLA-*b*-PGA-2. Another question that they should address is: Could it be that the apparent lower conversion of GA is due to the lack of solubility of PE-*b*-PEO-*b*-PCL-*b*-PLLA-*b*-PGA-2? To remove any doubts, the authors should therefore include a quantitative ¹H NMR spectrum to calculate the conversion of GA.

The authors should also include all the SEC traces for the synthesis of PE-*b*-PEO-*b*-PCL-*b*-PLLA-*b*-PGA-2 as well as all the solution states characterization for this polymer.

Answer: We could track any evidence of transesterification on PE-*b*-PEO-*b*-PCL-*b*-PLLA-*b*-PGA-2 because this sample is not soluble in any solvents/mix-solvents. Therefore, any solution-based analyses (¹H NMR and SEC) were not possible.

Due to the low solubility of the relatively high molecular weight PE block, and the insoluble nature of the PGA block, all attempts for dissolving pentablock-2 failed. 1,2,4-

trichlorobenzene (TCB), a common solvent for dissolving PE (also a common solvent for high temperature-SEC) cannot dissolve the PGA block at 150 °C, while hexafluoroisopropanol (*b.p.* 58.2 °C), a common fluorinated solvent for dissolving PGA, cannot dissolve PE block at 40 °C. Other common SEC solvents (e.g. THF, DMF, or CHCl₃) cannot dissolve either block.

As we mentioned in the main text, PE-*b*-PEO-*b*-PCL-*b*-PLLA-*b*-PGA-2 and PE-*b*-PEO-*b*-PCL-*b*-PLLA-*b*-PGA-1a were synthesized by the same method (with HFAB). In the main text, we have demonstrated that our method is excellent for obtaining pentacrystalline pentablock quintopolymer (pentablock-1a) with a low molecular weight PE precursor. With the same method, materials, and setups, we assume the pentablock-2 sample should also have a similar quality to pentablock-1a.

Addition to the MS: However, no solution-based analyses (¹H NMR and SEC) could be performed on pentablock-2 because the polymer is not soluble in any solvents/mix-solvents.

3) Also related to my first point, one of the biggest claims in the paper is that the fluoroalcohol-assisted catalyst switch is key for the synthesis of these pentablocks. However, I fail to see the advantage of using this method because in both cases, it looks like a pentablock with similar properties can be isolated. The authors should make a clear comparison between PE-*b*-PEO-*b*-PCL-*b*-PLLA-*b*-PGA-1a and PE-*b*-PEO-*b*-PCL-*b*-PLLA-*b*-PGA-2 to thoroughly explain why this strategy is essential.

Answer: The comparison should be made between PE-*b*-PEO-*b*-PCL-*b*-PLLA-*b*-PGA-1a (pentablock-1a) and PE-*b*-PEO-*b*-PCL-*b*-PLLA-*b*-PGA-1b (pentablock-1b). Pentablock-1a was synthesized in the presence of HFAB (as well as pentablock-2) whereas pentablock-1b was synthesized without HFAB as cosolvent.

The conversion of GA in pentablock-1b is significantly lower than that of pentablock-1a where HFAB is used. More importantly, this pentablock sample without the use of HFAB doesn't show a clear diffraction peak for the PGA block (22.5° and 29.2°), suggesting a weak crystallization behavior and cannot fulfill our main goal of achieving a pentacrystalline polymer. Moreover, HFAB can also facilitate the living/controlled ROP of

glycolide, which was described in detail in our recent paper (P. Zhang, V. Ladelta, and N. Hadjichristidis *J. Am. Chem. Soc.* 2023, 145, 27, 14756–14765).

Addition to the MS: Obviously, HFAB plays important roles in the solubilization of the PGA block and in facilitating the living/controlled ROP of GA.

4) A range of solution state 1D and 2D NMR spectroscopy techniques are used to characterized PE-b-PEO-b-PCL-b-PLLA-b-PGA-1a. The data presented by the authors are mostly convincing. However, there are four points that the author need to address:

a. First, the COSY spectrum in Fig. 2b is of overall low quality. The authors should process the spectrum via COSY symmetrisation and an arbitrary T1 noise reduction algorithm. The correlation of n->m can only be surmised, since the correlating symmetric signal is buried in the T1 noise streak of g+i+m. Adding to that, the authors should also improve the quality of the f2 trace, since the S/N ratio is high. There is also a visible cutoff on the f2 trace scale at approximately 0.5 ppm.

Answer: We reprocessed the COSY spectrum by following these step-by-step parameters:

- Apodization: Sine Square 0 for f2 & f1
- f2: 4096 datapoints
- f1: 2048 datapoints
- Phase Correction: automated
- Baseline Correction: Bernstein Polynomial Fit on either dimension
- Use COSY Symmetrization

After the above processing steps, the spectrum has become much more clear. We did not apply an arbitrary T1 noise reduction because it removed many important correlations from our spectrum. After reprocessing, two convincing correlations attributed to n->m and k->i are now well-resolved and can be observed clearly. The quality of the f2 trace is also improved.

Fig. 2b has been revised and placed in the main text.

Correction to the MS: The COSY spectrum has been revised.

Fig. 2 | Liquid-state NMR spectra of PE-*b*-PEO-*b*-PCL-*b*-PLLA-*b*-PGA-1a. **a.** 1D ^1H NMR spectra of pentablock-1a (Table 1, entries 1-5) and the corresponding precursors. **b.** ^1H - ^1H COSY spectrum of pentablock-1a (Table 1, entries 5). The experiments were carried out on a 600 MHz spectrometer in toluene- d_8 at 60 °C (except for PE-OH, 80 °C, toluene- d_8 , and the pentablock-1a, 60 °C, in a mixture of toluene- d_8 /HFIP- d_2 (~7/3 v/v)).

b. Second, in the ^1H - ^{13}C HMBC spectrum of PE-*b*-PEO-*b*-PCL-*b*-PLLA-*b*-PGA-1a (Fig. S15), one can see a cross peak at around 163 ppm aligned with methine peak of PLLA. Could the author comment on it? If it is a ghost peak, could the author comment on the risk that the peak they assigned as c->d, of similar relative intensity, is not also a ghost peak?

Answer: We reprocessed the ^1H - ^{13}C HMBC spectrum by following these step-by-step parameters:

- Apodization: Sine Square 90 for f2 & f1
- f2: 4096 datapoints
- f1: 2048 datapoints

- Phase Correction: automated
- Baseline Correction: Bernstein Polynomial Fit on either dimension

The cross peak at around 163 ppm aligned with the methine peak of PLLA is indeed a ghost peak, as it disappears after we reprocessed the signal on Mestrenova. However, the correlation $c \rightarrow d$ can be clearly observed, corroborating that this signal comes from a real correlation.

Revised Fig. S16 is given below:

Fig. S16. ^1H - ^{13}C HMBC spectrum of isolated PE-*b*-PEO-*b*-PCL-*b*-PLLA-*b*-PGA-1a (Table 1, entry 5, 600 MHz, 60 °C, toluene- d_8 and HFIP- d_2).

c. Third, the authors should describe in more details on the processing of the DOSY data. I refer them to this following paper: <https://doi.org/10.1002/macp.201900255> which show that a mixture of miscible homopolymers and block copolymers could lead to a single diffusion coefficient, if the data is not treated properly. They should ensure that their data treatment has been done properly.

Answer: Thanks for the comment. We reported the parameters for acquisition and processing for the DOSY experiment as follows:

Addition to the SI: The diffusion-ordered spectroscopy (DOSY) is performed on a Bruker 600 MHz liquid NMR spectrometer at 298.1 K. 32 gradient strengths varying linearly between 5% and 100% of the maximum gradient strength and 16 scans per increment were used to construct the decay function. D20 = 1.2000 s, P30 = 1500.00 μ s. The phase of the original data is adjusted manually and the baseline is smoothed via TopSpin 4.2.0. Bayesian method is used for processing the raw spectra. Resolution factor = 5.00 and repetitions = 3.

d. Fourth, the authors should do a quantitative analysis of the end-group conversion by using a phosphorous-containing reagent such as 2-chloro-4,4,5,5-tetramethyl dioxaphospholane to monitor the change in the chemical shift of the phosphorylated hydroxyl end group as a function of the growing of the next block. This would rule out any possibility of trans-esterification and the formation of other multiblock copolymers.

Answer: Thank you for the suggestion. In our Lab, we prefer to use trifluoroacetic anhydride (TFAA) instead of 2-chloro-4,4,5,5-tetramethyl dioxaphospholane as the reacting agent to monitor the possibility of trans-esterification (<https://doi.org/10.1016/j.polymer.2018.08.014>) because the procedure is straightforward, without the need to prepare a stock solution. We added 3 drops of TFAA into the NMR sample and performed ^{19}F NMR experiments on the tetrablock and pentablock solution. On the tetrablock spectrum, we could see the presence of a secondary –OH end group from the ^{19}F signal at ~ 77.1 ppm, which belongs to the PLLA end group. However, on the pentablock spectrum, we can not observe the existence of the PGA end group (primary –OH). This is because pentablock polymer requires fluorinated alcohol (HFAB) as a cosolvent and TFFA reacts with both –OH from HFAB and –OH from the PGA end-group. Therefore, end-group analysis using either phosphorylation or fluorination agents does not applicable to tracking transesterification reaction on pentablock sample.

Addition to the SI: New experiment

Fig. S13 ^{19}F NMR spectrum of TFAA in the presence of tetrablock-1 and pentablock-1a (hexafluorobenzene was used as internal standard). **a.** PE-*b*-PEO-*b*-PCL-*b*-PLLA-1 in toluene with 3 drops of TFAA. **b.** PE-*b*-PEO-*b*-PCL-*b*-PLLA-PGA-1a in toluene/ HFAB and 3 drops of TFAA. TFAA can react readily and thus can not be used to determine the end group of the pentablock (HFAB is needed for solvation).

Theoretically, there will be a small amount of transesterification during the polymerization of GA using $\text{Sn}(\text{Oct})_2$, particularly at high conversion. However, this transesterification tends to happen in the PGA chain only (no attack to the PLLA chain to create a secondary $-\text{OH}$ end group). This is also the reason we stopped the ROP of GA at $\sim 86\%$ conversion.

Addition to the MS: In order to further confirm the successful incorporation of the PLLA block, TFAA was added to detect the chain end. The appearance of a peak at -77.14 ppm on ^{19}F NMR spectrum after adding three drops of TFAA shows clearly the existence of the secondary hydroxyl group which belongs to the PLLA chain end $[-\text{CH}(\text{CH}_3)\text{OH}-\text{TFA}]$ (**Fig. S13**). After the ROP of GA, the signal for methylene group $-\text{C}=\text{OCH}_2\text{O}-$ was found at 4.53 ppm (*o*), confirming the successful synthesis of the pentablock quintopolymer. Further ^{19}F analysis to track any transesterification side reaction from PGA to PLLA block

could not be performed because TFAA reacted with HFAB (Fig. S13).

5) Related to the possibility of trans-esterification, the author mention in the discussion of the VT ^1H - ^{13}C CP-MAS NMR spectra that “a very weak signal at 17.7 ppm corresponding to PLLA also appears at 120 °C, indicating that a small fraction of PLLA starts to melt at this temperature”. They postulate that this could be do to “the mobility transfer from the melted amorphous block to the semi-crystalline PLLA block due to mechanical spinning”. The authors should demonstrate that this is not due to transesterification issues which would have shorten the PLLA chain in some of the polymer and would give similar results.

Answer: To prove this signal is not from short PLLA chains formed due to transesterification, we performed CP-MAS and INEPT on PLLA 1.4 K. The results clearly show that short PLA chains (~10 repeating units) do not give a signal on INEPT at 120 °C. This corroborates that mobility transfer affects the crystalline domain in the pentablock polymer. Please see the below figures.

Addition to the SI: New experiment

Fig. S33. a. ^1H - ^{13}C CP-MAS NMR spectrum and **b.** INEPT NMR spectrum of a 1.4 kg mol $^{-1}$ PLLA homopolymer at 120 °C.

Addition to the MS: A control experiment using low molecular weight PLLA (1.4 kg mol $^{-1}$)

1) homopolymer shows no INEPT signal at 120 °C (**Fig. S33**), excluding the possibility of forming oligomeric PLLA chains due to transesterification or other side reactions under such conditions.

6) The TGA results show significant change in the $T_{5\%}$ for PE-b-PEO-b-PCL-b-PLLA-b-PGA-1a and PE-b-PEO-b-PCL-b-PLLA-b-PGA-2 not only for the final polymers but also for each of the individual blocks. The authors should comment on these differences.

Answer: The significant differences in $T_{5\%}$ in each individual block are mainly due to the different mass percentages for each block. For pentablock-1a, the targeting molecular weight for each block is 1.5 kg mol⁻¹, 15 kg mol⁻¹, 15 kg mol⁻¹, 30 kg mol⁻¹, and 7 kg mol⁻¹. But for pentablock-2, the targeting molecular weight for each block is 7 kg mol⁻¹, 10 kg mol⁻¹, 10 kg mol⁻¹, 25 kg mol⁻¹, and 15 kg mol⁻¹, respectively. We targeted higher molecular weight for PEO, PCL, and PLLA (which have high solubility in common solvents) in pentablock-1a to increase the overall solubility of the pentablock-1a for clear characterizations (e.g. SEC and various NMR).

Please note that we repeated the TGA for pentablock-1a. We found that $T_{5\%}$ is around 199.9 °C. The previous result might be due to the residue catalyst accelerating the decomposition of the polymer.

Addition to the MS: Significant differences were observed in $T_{d,5\%}$ in pentablock-1a and pentablock-2 as well as for each individual block, which are mainly due to the different composition.

Minor Comments

7) The temperatures in the tables are difficult to read and the conversion in entry 5 in table S1 is missing even though the authors report a conversion in the text.

Answer: We changed the format of T_m/T_c and showed a better presentation. The conversion of GA in pentablock-2 is actually not obtained here as the sample was not fully soluble. We marked this point in the table footnote.

8) In the introduction, the authors wrote: “Recently, we developed a “fluoroalcohol-

assisted catalyst switch” strategy (P.Z., V.L., and N.H.) and reported the first living/controlled polymerization of glycolide at room temperature.” A reference is missing here. The authors should also remove or explain what the abbreviations in the brackets mean.

Answer: This is because the reference paper is not yet published online at the time of submission of this paper. We have removed the brackets and replaced them with a proper citation.

9) When the authors are discussing about the thermal properties of the different blocks, they should relate to their previous study and comment if similar (or different) behavior are observed.

Answer: In the main text, we have already explained the influence of adding the PGA block to the tetrablock precursor when discussing DSC (Figure 4) results. However, following the recommendations of the reviewer, we now include a paragraph to compare the results of the tetrablock precursor with a similar material synthesized and studied by us previously. In the case of pentablock polymers, we cannot compare them with any previous results, as this is the first time such a sophisticated material has been prepared.

The following text was added in the paper:

Addition to the MS: In our previous work⁵⁵, we investigated the thermal properties of a PE₁₈-*b*-PEO₃₇-*b*-PCL₂₆-*b*-PLLA₁₉ tetrablock quarterpolymer. Several advanced characterization techniques were employed to identify with high precision the crystallization sequence of the blocks upon cooling from the melt: PLLA first, then PE, then PCL, and finally PEO. The corresponding reverse sequence was observed during subsequent heating. A similar behavior was observed on tetrablock-1 as shown in Fig. 4, except for the magnitude of the first-order thermal transitions (and their scan rate dependence), whose enthalpies were different because of the different compositions and molecular weights in the two tetrablocks. In the future, we will perform *in situ* WAXS/SAXS

experiments at the synchrotron to determine precisely the sequential crystallization and melting of the pentablocks and their corresponding precursors.

Reviewers' Comments:

Reviewer #1:

Remarks to the Author:

thanks - the manuscript is excellent and ready to publish.

Reviewer #2:

Remarks to the Author:

In their rebuttal to my comments regarding the article "Catalyst Switch" Strategy Enabled Pentacrystalline Pentablock Quintopolymer: A Single Polymer with Five Different Crystalline Phases, the authors sufficiently answered our questions and incorporated additional arguments concerning the presented research, that improved the overall quality of the manuscript, making it suitable for publication.

However, I want to point out one discrepancy that I found in Fig. S13b in the provided ESI. The authors present the ^{19}F NMR spectrum of pentablock-1a in a mixture of toluene/HFAB with the addition of an excess of TFAA for end-group analysis. The denoted overlap of HFAB, trifluoroacetylated HFAB and end-group functionalized polymer seems against chemical intuition. I know that most trifluoroacetyl moieties appear around -75 ppm but an overlap of three different species at one precise value seems rather surprising. To remove any ambiguities, fluorine NMR spectra of HFAB as well as the reaction of HFAB with TFAA should be added. Alternatively, a reference should be added. The authors should also try to add an excess of TFAA compare to HFAB to see if they can see the signature from the end-group of the polymer.

Also, the authors wrote in P17 of their rebuttal: "Theoretically, there will be a small amount of transesterification during the polymerization of GA using $\text{Sn}(\text{Oct})_2$, particularly at high conversion. However, this transesterification tends to happen in the PGA chain only (no attack to the PLLA chain to create a secondary-OH end group)." They should add a reference for this claim or show experimental evidence.

Despite this small comments, I still recommend this manuscript for publication.

REVIEWERS' COMMENTS

Reviewer #1 (Remarks to the Author):

thanks - the manuscript is excellent and ready to publish.

Thank you for the reviewer's positive response

Reviewer #2 (Remarks to the Author):

In their rebuttal to my comments regarding the article "Catalyst Switch" Strategy Enabled Pentacrystalline Pentablock Quintopolymer: A Single Polymer with Five Different Crystalline Phases, the authors sufficiently answered our questions and incorporated additional arguments concerning the presented research, that improved the overall quality of the manuscript, making it suitable for publication.

However, I want to point out one discrepancy that I found in Fig. S13b in the provided ESI. The authors present the ^{19}F NMR spectrum of pentablock-1a in a mixture of toluene/HFAB with the addition of an excess of TFAA for end-group analysis. The denoted overlap of HFAB, trifluoroacetylated HFAB and end-group functionalized polymer seems against chemical intuition. I know that most trifluoroacetyl moieties appear around -75 ppm but an overlap of three different species at one precise value seems rather surprising. To remove any ambiguities, fluorine NMR spectra of HFAB as well as the reaction of HFAB with TFAA should be added. Alternatively, a reference should be added. The authors should also try to add an excess of TFAA compare to HFAB to see if they can see the signature from the end-group of the polymer.

Answer: As suggested by the reviewer, we performed ^{19}F NMR analyses of HFAB, TFAA, and HFAB mixed with a large excess of TFAA. The ^{19}F signals of HFAB, TFAA, and HFAB+TFAA are well separated (peaks m, n, and m'+n', Fig. S13). We also performed the end-group modification by adding 10 mL of TFAA to the tetrablock (dissolved in 4 mL dichloromethane) and pentablock (dissolved in a mixture of 4 mL dichloromethane + 0.1 mL HFAB [much less than TFAA]). The polymers were precipitated in methanol and dried *in vacuo* to remove the residual TFAA. The ^{19}F signals from trifluoroacetyl esters-derived pentablock and tetrablock (pentablock-TFA and tetrablock-TFA) can be observed at -76.16 ppm (k, corresponding to primary -OH) and -78.22 ppm (l, corresponding to secondary -OH), respectively. The order of the chemical shift positions is in good agreement with the literature (Wang et.al, Polymer, 2018, 153, 167-172), indicating that there is no transesterification side reaction caused by the attack of PGA living chain-end to the PLLA block.

Correction to the MS: In order to further confirm the successful incorporation of the PLLA block, TFAA was added to **modify the end-group**. The appearance of a peak **at -78.22 ppm** on the ^{19}F NMR spectrum after the **modification of the end-group** shows clearly the existence of the secondary hydroxyl group which belongs to the PLLA chain end [-CH(CH₃)OH-TFA] (**Supplementary Fig. 13**). After the ROP of GA, the signal for methylene group -C=OCH₂O- was found at 4.53 ppm (o), confirming the successful synthesis of the pentablock quintopolymer. **In addition, ^{19}F NMR analysis of the trifluoroacetyl ester-derived pentablock (pentablock-TFA) revealed the existence of only the primary hydroxyl group (Supplementary Fig. 13, ^{19}F signal at -76.16 ppm), indicating that there is no transesterification caused by the attack of PGA living chain-end to the PLLA block.**

Corrections to the SI: ^{19}F NMR Spectra.

Supplementary Figure 13 ^{19}F NMR spectra of: **a** HFAB, **b** TFAA, **c** HFAB reacted with a large excess of TFAA, **d** pentablock-TFA in the presence of HFAB, and **e** tetrablock-TFA (800 MHz, CDCl_3 , room temperature, hexafluorobenzene was used as internal standard). * signal corresponds to trifluoroacetic acid.

Also, the authors wrote in P17 of their rebuttal: "Theoretically, there will be a small amount of transesterification during the polymerization of GA using $\text{Sn}(\text{Oct})_2$, particularly at high conversion. However, this transesterification tends to happen in the PGA chain only (no attack to the PLLA chain to create a secondary-OH end group)." They should add a reference for this claim or show experimental evidence.

Answer: If there is any attack from the living PGA chain to the PLLA block, lactyl-glycolyl or glycolyl-lactyl (LG or GL) units will be formed and distributed randomly along the chain. As a consequence, the ^{13}C signals of PLLA and PGA (Fig S17, peaks: i, j, k, l, and m) will split due to the unique chemical environment (Ryan M. Stayshich and Tara Y. Meyer, *J. am. chem. soc.* 2010,132, 10920–10934). However, ^{13}C signals of i, j, k, l, and m of the pentablock (Fig. S17) appear as singlet peaks (150 MHz, 19200 scans), indicating that there is no transesterification caused by the attack of PGA living chain-end to the PLLA block.

Addition to the MS: Furthermore, the signals of $-\text{C}=\text{O}$, $-(\text{C})\text{H}$, $-\text{CH}_2-$, and $-\text{CH}_3$ of PLLA and PGA (i, j, k, l, and m) appear as singlet peaks, corroborating that there is no lactyl-glycolyl or glycolyl-lactyl units formed due to transesterification from the attack of PGA living chain end to the PLLA block.

Corrections: We added three insets to Supplementary Fig. 17.

Supplementary Figure 17. ¹³C NMR spectrum of isolated PE-*b*-PEO-*b*-PCL-*b*-PLLA-*b*-PGA-1a (Table 1, entry 5, 150 MHz, 60 °C, toluene-*d*₈ and HFIP-*d*₂).

Despite this small comments, I still recommend this manuscript for publication.
Thank you for the reviewer's positive response